# AFTER: Mitigating the Object Hallucination of LVLM via Adaptive Factual-Guided Activation Editing

**Tianbo Wang**[1,3], **Yuqing Ma**[2,3,*], **Kewei Liao**[1,3], **Zhange Zhang**[2,3], **Simin Li**[1,3], **Jinyang Guo**[2,3], **Xianglong Liu**[1,3]

[1]School of Computer Science and Engineering, Beihang University
[2]Institute of Artificial Intelligence, Beihang University
[3]State Key Laboratory of Complex & Critical Software Environment
{tianbowang, mayuqing}@buaa.edu.cn

## Abstract

Large Vision-Language Models (LVLMs) have achieved substantial progress in cross-modal tasks. However, due to language bias, LVLMs are susceptible to object hallucination, which can be primarily divided into category, attribute, and relation hallucination, significantly impeding the trustworthy AI applications. Editing the internal activations of LVLMs has shown promising effectiveness in mitigating hallucinations with minimal cost. However, previous editing approaches neglect the positive guidance offered by factual textual semantics, thereby struggling to explicitly mitigate language bias. To address these issues, we propose **A**daptive **F**actual-guided Visual-**T**extual **E**diting fo**R** hallucination mitigation (AFTER), which comprises Factual-Augmented Activation Steering (FAS) and Query-Adaptive Offset Optimization (QAO), to adaptively guide the original biased activations towards factual semantics. Specifically, FAS is proposed to provide factual and general guidance for activation editing, thereby explicitly modeling the precise visual-textual associations. Subsequently, QAO introduces a query-aware offset estimator to establish query-specific editing from the general steering vector, enhancing the diversity and granularity of editing. Extensive experiments on standard hallucination benchmarks across three widely adopted LVLMs validate the efficacy of the proposed AFTER, notably achieving up to a 16.3% reduction of hallucination over baseline on the AMBER benchmark. [1]

## 1 Introduction

Building upon the foundation of Large Language Models (LLMs), Large Vision-Language Models (LVLMs) have made substantial advancements in cross-modal understanding and generation (Bai et al., 2023; Ye et al., 2024). However, LVLMs continue to grapple with a significant challenge known as *object hallucination* (Bai et al., 2024; Liu et al., 2024b), which refers to discrepancies between the factual visual objects and the model-generated response. This issue severely impedes the trustworthiness of LVLMs in real-world applications (Yan et al., 2024; Xie et al., 2025).

Existing studies have demonstrated that one primary cause of hallucination is the language bias (Bai et al., 2024; Jiang et al., 2024b; Liu et al., 2024a), which leads LVLM to prioritize textual knowledge over the external visual inputs. As illustrated in Figure 1, language bias empirically results in three primary types of hallucination (Bai et al., 2024; Liu et al., 2024b): (1) *Category Hallucination*: The object category "backpack" is mistakenly identified as a "snowboard" due to the language prior associating skiing with snowboards (Niu et al., 2021). (2) *Attribute Hallucination*: The incorrect object attribute (*e.g.* counting) of gloves arises from the prior that gloves typically appear in pairs (Niu et al., 2021; Agrawal et al., 2018). (3) *Relation Hallucination*: The frequent prior "man wearing a helmet" overrides the object relation fact "man holding a helmet" (Agrawal et al., 2018). Although

---

[*]Corresponding author.
[1]Our code is available at https://github.com/wytbwytb/AFTER.

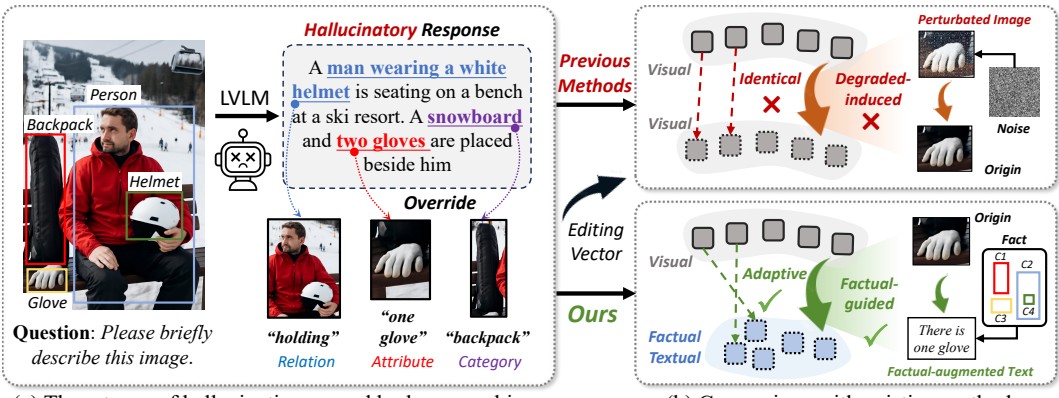

Figure 1: (a) Demonstration of the three types of hallucinations (category, attribute, and relation) caused by language bias. (b) Comparisons between previous activation editing methods and AFTER.

existing hallucination mitigation methods, *e.g.* training-based (Ouali et al., 2024; Wang et al., 2024a) and inference-time (Chen et al., 2024a; Zhang et al., 2025), have gained notable success, their applications are constrained by excessive training burden or multi-round inference costs (Chen et al., 2025).

Recently, inference-time activation editing techniques (Liao et al., 2025; Chen et al., 2024b; Qiu et al., 2024; Wang et al., 2025; Zhang et al., 2024) have shown promise in addressing hallucinations in LVLMs (Chen et al., 2025; Liu et al., 2025). Through employing carefully designed editing vectors, these techniques can directly optimize LVLMs' behavior by editing the hallucinatory internal activation with minimal inference costs. For instance, VTI (Liu et al., 2025) constructs the vector by contrasting stable visual features (averaged from multiple perturbed images) with the original ones, then applies interventions in the visual encoder to enhance activation stability. ICT (Chen et al., 2025) generates globally noisy and locally blurred images as untrusted semantics, which are used to calculate separate editing vectors to improve the comprehension of image information and object details in LVLMs, respectively.

However, although prior methods intentionally degrade visual semantics (*e.g.* injecting perturbations into images) to steer activations within the visual space, they overlook the positive guidance offered by factual textual semantics. As a result, these methods fail to capture diverse visual-textual associations, limiting their ability to explicitly mitigate language bias. Specifically, the factual information embedded in the image's ground-truth annotations cannot be textualized by existing methods to construct positive steering directions, thereby failing to tackle visual-textual disparity (Jiang et al., 2024a; Sun et al., 2024). Additionally, the diverse query-emphasized objects exhibit distinct visual-textual associations with specific offsets from the general one, which existing identical steering vectors cannot accommodate.

Therefore, we propose **A**daptive **F**actual-Guided Visual-**T**extual **E**diting fo**R** hallucination mitigation (**AFTER**), which comprises *Factual-Augmented Activation Steering (FAS)* and *Query-Adaptive Offset Optimization (QAO)*, to adaptively steer original activation toward factual-augmented textual semantics for language bias alleviation. FAS first leverages factual information to provide positive and explicit textual guidance for visual-textual activation editing. It innovatively transforms ground-truth annotations into textual category, attribute, and relation facts, thereby generating trusted text-query samples that are resistant to language bias. Subsequently, FAS can derive a general and positive visual-textual steering direction by contrasting trusted textual activations with original activations, thereby effectively guiding the activations to tackle visual-textual disparity. To further promote editing diversity, QAO introduces a query-aware offset estimator to assess distinct deviations from the general steering vector, therefore establishing query-specific visual-textual associations. QAO specifically evaluates the overlap between query-referenced objects and entire category facts to generate query-specific offsets. This guides the estimator to adaptively steer LVLMs towards prioritizing edited visual semantics, thereby mitigating language bias. We summarize our contributions as follows:

- We propose the AFTER, an effective activation editing approach to adaptively steer original activation toward factual-augmented semantics for hallucination mitigation.

- We introduce Factual-Augmented Activation Steering (FAS), which leverages factual textual semantics to provide positive guidance for activation editing of LVLM.

- We propose Query-Adaptive Offset Optimization (QAO), which further establishes query-specific visual-textual association based on the general vector to promote diversity.

- Extensive experiments reveal that our method achieves superior performance with minimal cost, outperforming baselines by up to 16.3% reduction on AMBER. It also exhibits strong generalizability and proves effective in enhancing common visual-textual capability.

## 2 Related Works

### 2.1 Large Vision-Language Models

Building on the successful application of Large Language Models (LLMs), Large Vision-Language Models (LVLMs) enhance the visual perception of LLMs (Touvron et al., 2023; Chiang et al., 2023) by integrating a pre-trained visual encoder (Radford et al., 2021; Fang et al., 2023), achieving significant performance in diverse vision-language tasks (Plummer et al., 2015; Chen et al., 2015; Schwenk et al., 2022; Hudson & Manning, 2019). To establish the connection between visual and textual representation, LVLMs usually incorporate a learnable interface, which can be broadly classified into query-based and projection-based (Bai et al., 2024; Jiang et al., 2024a). Query-based methods, such as InstructBLIP (Dai et al., 2023), MiniGPT-4 (Zhu et al., 2024) with Q-Former, utilize a set of learnable query tokens to capture visual signals via cross-attention. Represented by LLaVA (Liu et al., 2023) and Shikra (Chen et al., 2023), projection-based methods utilize a trainable linear projection layer or a Multi-Layer Perceptron (MLP) to transform extracted visual features. In this work, we selected three commonly used LVLMs of LLaVA-v1.5, Shikra, and InstructBLIP to evaluate our approach.

### 2.2 Hallucination Mitigation of LVLM

Current LVLM hallucination mitigation methods fall into training-based and inference-time approaches. Training-based methods retrain LVLMs with high-quality data (Liu et al., 2024a; Yu et al., 2024; Ouali et al., 2024) or new objectives (Jiang et al., 2024a; Lyu et al., 2024), but are time-consuming and resource-intensive. Inference-time methods mitigate hallucinations during generation via specialized decoding (Leng et al., 2024; Huang et al., 2024; Chen et al., 2024a) or iterative corrections (Lee et al., 2024; Yin et al., 2024), but require multiple inference steps that increase inference cost. Currently, several works (Liu et al., 2025; Chen et al., 2025) have demonstrated that directly editing the internal activations of LVLM during inference can mitigate hallucination. For example, VTI (Liu et al., 2025) constructs a vector by contrasting stable visual features (averaged from perturbed images) with the original ones, then applies interventions in the visual encoder to enhance activation stability. ICT (Chen et al., 2025) generates globally noisy and locally blurred images as untrusted semantics, computing separate editing vectors to improve the comprehension of image information and object details in LVLMs. Compared with previous model-specific methods, it also incurs relatively lowest overhead, offering practical transferability. However, they fail to capture the query-specific visual-textual association, thereby limited to explicitly mitigate language bias.

## 3 Methodology

To effectively reduce query-specific language bias, we introduce Adaptive Factual-guided Visual-Textual Editing foR hallucination mitigation (AFTER). AFTER initially leverages Factual-Augmented Activation Steering (FAS) to establish the general and truthful visual-textual editing direction, thereby steering original hallucinatory activation toward factual-guided textual semantics. Subsequently, Query-Adaptive Offset Optimization (QAO) is introduced to generate necessary offset on the general vector, enabling adaptive and precise editing for distinct queries. In this section, we first present preliminary in Section 3.1, and elaborate on FAS in Section 3.2 and QAO in Section 3.3.

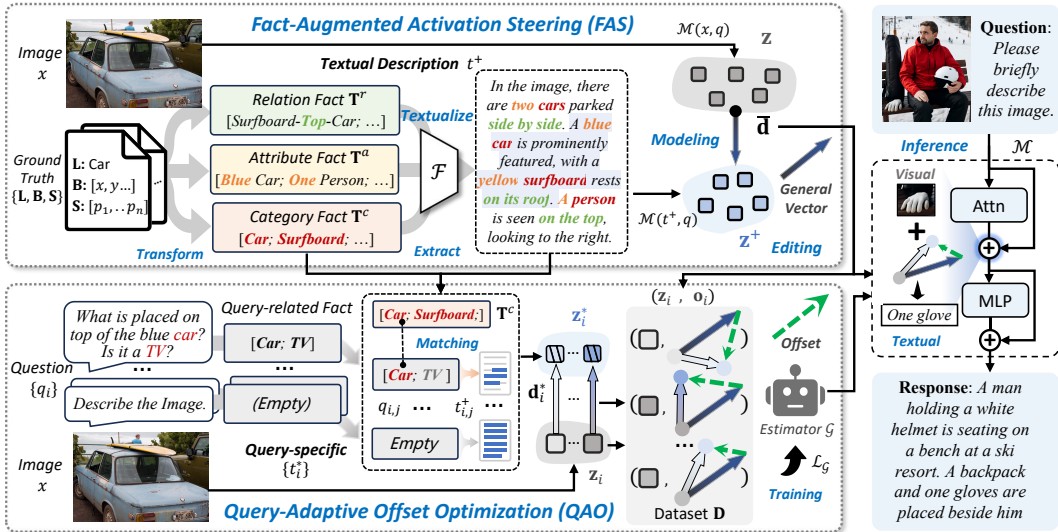

Figure 2: An overview of the AFTER. FAS first establishes the general and positive visual-textual editing direction with the guidance of facts. QAO then achieves precise query-adaptive editing by training a query-aware offset estimator, thereby explicitly mitigating language bias.

## 3.1 PRELIMINARY

Given an LVLM $\mathcal{M}$ encoded with rich pretrained language knowledge, the model can process a query composed of an image-question pair $\langle x, q \rangle$, and generate an answer $y = \mathcal{M}(x, q)$. During forward of $\mathcal{M}$, the image-question pair $\langle x, q \rangle$ is tokenized and subsequently passed through $L$ decoding layers with $H$-head self-attention, yielding the hidden states at each layer as $\mathbf{h}^l$:

$$\mathbf{h}^{l+1} = \mathbf{h}^l + \text{Concat}_{k=1}^{H}(\mathbf{z}^{l,k}) \cdot W_o^l, \tag{1}$$

where $\mathbf{z}^{l,k} = \text{Attn}^{l,k}(\mathbf{h}^l)$ denotes the internal activation after self-attention operation of the $k$-th head at the $l$-th layer, $W_o^l$ is an output projection matrix. However, $\mathcal{M}$ tends to prioritize textual knowledge over the external visual input $x$ due to language bias, rendering the generated answer $y$ to be hallucinatory. Therefore, sparse interventions on the internal activations have been designed by activation editing to guide the model toward producing non-hallucinatory outputs.

Typically, these methods first construct steering vector $\bar{\mathbf{d}} = \sum_{\mathbf{X}}(\mathbf{z}^+ - \mathbf{z}^-)/|\mathbf{X}|$ by averaging the differences between trusted visual activation $\mathbf{z}^{+}$ [2] and untrusted visual activation $\mathbf{z}^-$ across image set $\mathbf{X}$. The editing vector $\bar{\mathbf{d}}$ is then applied to the internal activation during inference as follows:

$$\mathbf{h}^{l+1} = \mathbf{h}^l + \text{Concat}_{k=1}^{H}(\mathbf{z}^{l,k} + \alpha \cdot \bar{\mathbf{d}}) \cdot W_o^l, \tag{2}$$

where $\alpha$ denotes the editing intensity. However, previous methods typically degrade image $x$ to obtain trusted activation $\mathbf{z}^+$ and untrusted activation $\mathbf{z}^-$, failing to establish factual steering guidance. In contrast, FAS in Section 3.2 augment $x$ with abundant facts to generate factual textual description $t^+$, thereby providing positive guidance by extracting $\mathbf{z}^+$ from factual $(t^+, q)$ and $\mathbf{z}^-$ from original $(x, q)$. Additionally, prior researches ignore query-specific visual-textual associations and employ identical averaged vectors $\bar{\mathbf{d}}$ for editing. QAO in Section 3.3 specially estimates the query-specific offset $\mathbf{o}_i$ based on the averaged vector $\bar{\mathbf{d}}$, realizing query-adaptive factual-guided activation editing.

## 3.2 FACTUAL-AUGMENTED ACTIVATION STEERING

To fully exploit factual textual semantics for positive editing guidance, we propose Factual-Augmented Activation Steering (FAS) to directly reduce language bias. FAS intuitively treats the original visual information as untrusted semantics, and our fact-augmented textual description as trusted semantics,

---

[2]Due to the identical operation, we omit the layer $l$ and head $k$ indices in the upper right corner for all activation symbols in following Sections to simplify the notation.

therefore explicitly constructing reliable visual-textual editing vectors. This enables positive steering of the original hallucinatory activation, thereby preventing the misguidance of language bias.

To facilitate the generation of factual textual descriptions as trusted semantics, we innovatively textualize the ground-truth annotations into category, attribute, and relation facts, thereby effectively mitigating the three types of hallucination. Specifically, we sample an image set $\mathbf{X}$ from the classic COCO (Lin et al., 2014) training set, each image $x \in \mathbf{X}$ accompanied by rich ground-truth annotations of core objects. The transformations of ground-truth annotations into category fact set $\mathbf{T}^c$, attribute fact set $\mathbf{T}^a$, and relation fact set $\mathbf{T}^r$ are illustrated as follows (More details are presented in Appendix C):

- **Category fact set $\mathbf{T}^c$**: The category facts correspond to the factual description of object categories, which can be generated by directly integrating the category labels $\mathbf{L}$ of all objects.
- **Attribute fact set $\mathbf{T}^a$**: In the attribute fact set $\mathbf{T}^a$, the focused facts primarily include color, shape, and count:
  - **Color**: The color attribute is manually annotated based on pixel-level statistics within the objects. We specifically designate the color with the highest pixel proportion in segmented region as the object's color attribute.
  - **Shape**: This attribute refers to the objects' shape (*e.g.* circular, square), which are transformed from the segmentation polygons $\mathbf{S}$ by approximating their contours and analyzing geometric regularities such as vertex count and angular consistency.
  - **Count**: The count attribute denotes the occurrence frequency of a particular category within the image, which can be calculated according to category labels $\mathbf{L}$.
- **Relation fact set $\mathbf{T}^r$**: Relation facts can be estimated from the spatial relationships (*e.g.* left, overlapped) between bounding boxes annotations $\mathbf{B}$. This process is achieved by computing the directional offsets between the box centers and spatial proximity according to their IoU.

After accurately extracting the three types of hallucination-related facts, we textualize all the facts into a comprehensive and factual description with the help of existing LVLM:

$$t^+ = \mathcal{F}(\mathrm{I_{fst}}; (x, [\mathbf{T}^c, \mathbf{T}^a, \mathbf{T}^r])), \tag{3}$$

where $t^+$ denotes the textualized factual description by LVLM $\mathcal{F}$ with instruction $\mathrm{I_{fst}}$ (shown in Appendix F). It is worth noting that $\mathcal{F}$ is employed solely for integrating discrete facts into coherent textual ground-truth, which is necessary for editing methods (Li et al., 2024a), without providing extra information. The capabilities of $\mathcal{F}$ are not engaged during the inference of the edited model $\mathcal{M}$, thereby ensuring a fair comparison with other methods.

Subsequently, FAS can construct trusted-untrusted sample pairs $\langle (t^+, q), (x, q) \rangle$ by concatenating trusted textual description $t^+$ and untrusted visual images $x$ with question $q$, facilitating the modeling of positive editing directions. Specifically, for each image $x$ and corresponding textual description $t^+$, we construct an $n$-question set $\{q_i\}$ associated with diverse object facts, where each question $q_i$ (*e.g.*, *Describe this image.*) has the potential to elicit a hallucinatory response. Subsequently, we combine visual image $x$ and textual description $t^+$ with every generated question, forming $n$ trusted-untrusted sample pairs $\{\langle (t^+, q_i), (x, q_i) \rangle | i \in [1, n]\}$. The samples are then input into LVLM $\mathcal{M}$ to obtain the trusted-untrusted activation pairs $\langle \mathbf{z}_i^+, \mathbf{z}_i \rangle$, which represent the factual textual semantics and original hallucinatory semantics perceived by $\mathcal{M}$, respectively. Therefore, we can directly model the general visual-textual steering vector by averaging the differences between $\mathbf{z}_i^+$ and $\mathbf{z}_i$ across the whole image set $\mathbf{X}$, which is a common practice for activation editing (Chen et al., 2025; Li et al., 2024b):

$$\bar{\mathbf{d}} = \frac{1}{n \cdot |\mathbf{X}|} \sum_{\mathbf{X}} \sum_{i=1}^{n} (\mathbf{z}_i^+ - \mathbf{z}_i), \tag{4}$$

where $\bar{\mathbf{d}}$ denotes the general visual-textual editing vector, $|\mathbf{X}|$ denotes the number of calculated images. Therefore, FAS can explicitly reduce the language bias by applying the general steering vector to perform beneficial editing, thereby mitigating the hallucinatory response.

### 3.3 QUERY-ADAPTIVE OFFSET OPTIMIZATION

Distinct visual semantics emphasized by different queries require specialized editing to more precisely reduce language bias. This motivates the need to apply an adaptive offset on the general visual-textual

vector, thereby constructing steering vectors tailored to the specific query. To this end, we propose Query-Adaptive Offset Optimization (QAO), which intuitively devises a query-aware offset estimator that fully captures query-relevant visual semantics and estimates the necessary offset accordingly.

To provide a specific data foundation for training the offset estimator, we first generate more detailed textual descriptions of query-emphasized visual semantics. Specifically, given image $x$, its textual description $t^+$, and a question $q_i$, we first extract all object categories $\{q_{i,j}\}$ mentioned in $q_i$, which constitute the query-relevant visual details that LVLM is expected to attend to. Therefore, we seek to obtain object-related textual description $t_{i,j}^+$ of each $q_{i,j}$ according to the following principles:

$$t_{i,j}^+ = \begin{cases} \mathcal{F}(\mathrm{I_{qst}}; t^+, q_{i,j}) & , q_{i,j} \in \mathbf{T}^c \\ \text{``There is no } [q_{i,j}] \text{ in the image."} & , q_{i,j} \notin \mathbf{T}^c \end{cases} \tag{5}$$

This process means that if the query-related object is present in the image (*i.e.* $q_{i,j} \in \mathbf{T}^c$), we prompt $\mathcal{F}$ with instruction $\mathrm{I_{qst}}$ to extract the corresponding sub-description related to $q_{i,j}$ from the whole textual description $t^+$. Otherwise, we explicitly describe that the queried object is not present in the image. It is noticed that if $q_{i,j}$ does not mention any object (*e.g. Please describe this image.*), the original textual description $t^+$ is retained. We ultimately obtain the query-focused textual factual description $t_i^* = [t_{i,j}^+]_{j=1}^m$, where $[\cdot]$ denotes the concatenation of $m$ textual segments.

Upon obtaining the query-emphasized textual description, we are able to construct query-focused trusted-untrusted sample pairs $\langle (t_i^*, q_i), (x, q_i) \rangle$, and extract corresponding trusted-untrusted activation pairs $\langle \mathbf{z}_i^*, \mathbf{z}_i \rangle$. The precise query-specific disparity $\tilde{\mathbf{d}}_i = \mathbf{z}_i^* - \mathbf{z}_i$ serves as the optimal editing vector for the current query. Therefore, aiming to estimate the necessary offset needs to be added on the general vector $\bar{\mathbf{d}}$, we construct a training dataset $\mathbf{d} = \{(\mathbf{z}_i, \mathbf{o}_i) | i \in [1, n]\}$, where $\mathbf{o}_i = \tilde{\mathbf{d}}_i - \bar{\mathbf{d}}$ denotes the expected offset. Based on $\mathbf{D}$, we train the offset estimator $\mathcal{G}$ to comprehend the query-focused visual semantics $\mathbf{z}_i$ and estimate the offset $\mathbf{o}_i$ between the query-specific vector $\tilde{\mathbf{d}}_i$ and the general vector $\bar{\mathbf{d}}$. During training, we adopt the Mean-Square Error (MSE) loss to measure the discrepancy between the estimated offset and the expected offset:

$$\mathcal{L}_\mathcal{G} = \frac{1}{n \cdot |\mathbf{X}|} \sum_{\mathbf{X}} \sum_{i=1}^n \|\mathcal{G}(\mathbf{z}_i) - \mathbf{o}_i\|^2. \tag{6}$$

Thus, we can obtain the optimized editing vector for steering the query-focused activation towards factual textual semantics. It is worth noting that training $\mathcal{G}$ is highly efficient, as it is both lightweight (single-layer MLP) and does not require fine-tuning of LVLM. More experimental statistics can be seen in Appendix A.2. Ultimately, we directly apply query-guided editing to the top-$K$ heads most affected by language bias (*i.e.* those exhibiting the largest vector magnitudes). The adaptive visual-textual editing can be formulated as:

$$\mathbf{h}^{l+1} = \mathbf{h}^l + \mathrm{Concat}_{k=1}^H (\mathbf{z}^{l,k} + \alpha \cdot [\mathcal{G}(\mathbf{z}^{l,k}) + \bar{\mathbf{d}}]) \cdot W_o^l, \tag{7}$$

where $\alpha$ denotes the editing intensity. Through query-adaptive factual-guided editing, the LVLM allocates greater attention to the post-edited visual information, thereby mitigating hallucination.

## 4 EXPERIMENTS

### 4.1 EXPERIMENTAL SETUP

**Benchmarks and Metrics** We assess the performance of LVLMs under both discriminative and generative tasks. **For *discriminative* task**, we use the widely adopted POPE (Li et al., 2023b) and MME (Fu et al., 2023) to evaluate diverse types of hallucinations. Following (Leng et al., 2024; Chen et al., 2025), we compare different methods on the POPE task and report the average Accuracy and F1-score across the three datasets (COCO (Lin et al., 2014), A-OKVQA (Schwenk et al., 2022), GQA (Hudson & Manning, 2019)) and three settings (random, popular, and adversarial). On MME benchmark that evaluates general capabilities as well as object hallucination (including existence, count, position, and color), we adopt the MME score as the comprehensive metric to provide a quantitative measure. **For *generative* task**, we employ the generative subset of AMBER (Wang et al., 2023), which assesses the generative hallucination using metrics CHAIR (Rohrbach et al., 2018) and Hal. It also incorporates metric Cover to quantify the comprehensiveness of the response.

| Models | Methods | POPE | | MME | | | | AMBER | | |
|---|---|---|---|---|---|---|---|---|---|---|
| | | ACC(↑) | F1(↑) | E(↑) | CT(↑) | P(↑) | CR(↑) | CH(↓) | Hal(↓) | Cr(↑) |
| LLaVA-v1.5 | Baseline | 80.1 | 82.3 | 180.0 | 158.3 | 123.3 | 155.0 | 6.9 | 31.6 | 48.9 |
| | HACL | 83.5 | 83.0 | 185.0 | **168.3** | 133.3 | 145.0 | 7.1 | 31.4 | **49.6** |
| | VCD | 82.5 | 82.7 | 190.0 | 148.3 | 126.7 | 158.3 | 5.1 | 27.6 | 48.6 |
| | OPERA | 83.3 | 83.5 | 190.0 | 153.3 | 123.7 | 158.3 | 4.9 | 27.9 | 49.0 |
| | VTI | 83.2 | 83.4 | 185.0 | 163.3 | 128.3 | 150.0 | 5.1 | 23.7 | 47.8 |
| | ICT | 83.7 | 83.7 | **195.0** | 158.3 | 126.7 | 158.3 | 5.4 | 26.6 | 48.8 |
| | *w/o* QAO | 83.8 | 84.4 | **195.0** | 163.3 | 128.3 | 160.0 | 5.2 | 22.3 | 48.6 |
| | Ours | **85.7** | **85.6** | **195.0** | 163.3 | **138.3** | **165.0** | **4.5** | **20.5** | 48.7 |
| Instruct-BLIP | Baseline | 80.3 | 82.0 | 175.0 | 60.0 | 50.0 | 120.0 | 7.4 | 35.4 | 53.5 |
| | VCD | 81.5 | 82.1 | 180.0 | 60.0 | 48.3 | 125.0 | 6.9 | 32.3 | **53.8** |
| | OPERA | 82.0 | 82.3 | 180.0 | 65.0 | 58.3 | 128.3 | 6.6 | 31.4 | 53.5 |
| | VTI | 82.3 | 82.7 | 170.0 | 60.0 | 53.3 | 120.0 | 5.3 | 26.7 | 53.0 |
| | ICT | 82.6 | 82.9 | 180.0 | 60.0 | 56.7 | 130.0 | 6.2 | 30.8 | 53.6 |
| | *w/o* QAO | 82.9 | 83.8 | **185.0** | 65.0 | 53.3 | 128.3 | 5.8 | 28.6 | 53.7 |
| | Ours | **83.5** | **84.2** | **185.0** | **70.0** | **63.3** | **133.3** | **5.2** | **25.1** | 53.6 |
| Shikra | Baseline | 78.9 | 80.3 | 185.0 | 66.7 | 58.3 | 103.3 | 10.9 | 49.5 | 50.7 |
| | VCD | 80.2 | 81.2 | 185.0 | 86.7 | 60.0 | 96.7 | 9.7 | 46.9 | 50.2 |
| | OPERA | 80.2 | 81.1 | 185.0 | 85.0 | 63.3 | 106.7 | 8.9 | 42.8 | **51.0** |
| | VTI | 80.6 | 81.3 | 185.0 | 83.3 | 55.0 | 101.7 | 7.5 | 38.5 | 48.6 |
| | ICT | 80.9 | 81.6 | **190.0** | 95.0 | 61.7 | 103.7 | 8.7 | 42.5 | 50.8 |
| | *w/o* QAO | 81.1 | 81.6 | **190.0** | 106.7 | **66.7** | 103.7 | 7.9 | 38.2 | 50.6 |
| | Ours | **82.5** | **82.5** | **190.0** | **116.7** | **66.7** | **113.3** | **6.9** | **33.2** | 50.4 |

Table 1: Comparison of AFTER with SOTA methods on POPE, MME, and AMBER. *w/o* **QAO** denotes our AFTER excluding QAO. The best results are in **bold**. Each result is reported under multiple rounds. The short names in MME correspond with four hallucinations mentioned in Section 4.1. The short names "CH" and "Cr" refer to the metric CHAIR and Cover for AMBER, respectively.

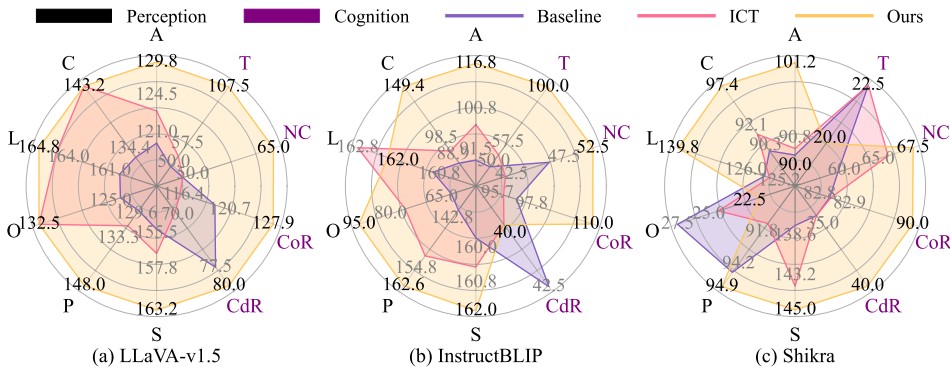

Figure 3: Comparison of AFTER with SOTA editing methods on MME's other capabilities. The short names for all capabilities are illustrated in Appendix B.

**Baseline and Comparative Methods** We choose three commonly-used LVLMs, including LLaVA-v1.5 (Liu et al., 2024c), InstructBLIP (Dai et al., 2023), and Shikra (Chen et al., 2023) as baselines. To evaluate our superiority, we first compare AFTER with existing activation editing methods, *i.e.* VTI (Liu et al., 2025) and ICT (Chen et al., 2025). We also consider other typical inference-time decoding-based methods, including VCD (Leng et al., 2024) and OPERA (Huang et al., 2024). Additionally, the training-based method HACL (Jiang et al., 2024a) is involved for comparison.

**Implementation Details** We randomly sample 500 images from the COCO training set to model visual-textual steering vector. For each image, we generate 6 questions following the protocols of POPE and obtain 1 question as AMBER to construct trusted-untrusted sample pairs. We try GPT-4o, GPT-4o-mini, and LLaVA-v1.5 to generate $t^+$. We adhere to the experimental setup outlined in (Leng et al., 2024; Chen et al., 2025) for fair comparison. Without specifying, the edited heads num $K$ is set to 64, and the editing strength $\alpha$ is set to 7. More details are provided in Appendix C.

## 4.2 EXPERIMENTAL RESULTS

**Hallucination Mitigation Performance**   Table 1 shows the comparison between AFTER and various hallucination mitigation methods on POPE, MME, and AMBER to illustrate our effectiveness on both discriminative and generative tasks.

Obviously, our method **demonstrates *discriminative* advantages** in both POPE and MME benchmarks across three prevailing LVLMs. On POPE, we achieve an average improvement of 4.1% in accuracy and 2.6% in F1-score over the baselines, surpassing the SOTA editing method ICT by 1.3% and 0.9%. Additionally, on the hallucination subset of MME (Zhuang et al., 2025; Chen et al., 2025), AFTER yields score improvements of 45.0, 46.6, and 73.4 on LLaVA-v1.5, InstructBLIP, and Shikra compared to the vallina LVLM, outperforming all SOTA methods. This enhancement demonstrates the superiority of the adaptive factual-guided visual-textual editing of AFTER, which effectively avoids the misguidance of language bias by steering original hallucinatory activation towards factual textual semantics.

We also achieve the **optimal *generative* hallucination mitigation** on AMBER, with an averaged 2.9% and 12.6% reduction on CHAIR and Hal metrics over the baselines. When applied to Shikra, we particularly reduce the hallucination by 16.3%, superior to the suboptimal editing method VTI by 5.3%. Therefore, without compromising the LVLM's comprehensive understanding of images (negligible change in the Cover metric, a trade-off with hallucination suppression (Han et al., 2025)), AFTER effectively reduces hallucinated objects during generation by leveraging factual visual-textual guidance. It is noticed that solely deploying the factual-guided vector for editing will bring slightly lower improvement on the three benchmarks. This manifests that query-adaptive editing with the guidance of QAO is also essential for precisely reducing query-specific language bias.

**Foundational Visual-language Performance**   As indicated in Figure 3, we also exceed the baseline model and best editing method ICT on almost every dimension that evaluates the general visual perception and cognition capabilities, with an average of 130.7 increased score on three LVLMs. These results indicate that our AFTER not only effectively reduces hallucinations but also enhances general visual capabilities across different models, which benefits from the superiority of steering the visual activation toward factual-guided textual semantics adaptively to alleviate language bias.

**Generalization Performance**   We also evaluate the generalizability of AFTER by directly applying the factual visual-textual steering vectors learned from COCO-based discriminative questions to out-of-distribution benchmarks. Following (Chen et al., 2025), we generalize these vectors on GQA-based POPE evaluation (COCO → GQA, with different class space) and generative AMBER benchmark (Dis → Gen) to estimate the generalization performance across visual images and textual questions, respec-

| Models | Methods | COCO → GQA | | Dis → Gen | |
| | | ACC | F1 | Hal | Cover |
|---|---|---|---|---|---|
| LLaVA-v1.5 | Baseline | 76.9 | 80.3 | 31.6 | **48.9** |
| | Ours | **84.6** | **84.8** | **22.8** | 48.7 |
| Instruct-BLIP | Baseline | 77.9 | 80.5 | 35.4 | 53.5 |
| | Ours | **81.4** | **82.4** | **27.9** | **53.8** |
| Shikra | Baseline | 78.4 | 80.0 | 49.5 | 50.7 |
| | Ours | **82.3** | **82.5** | **38.5** | **51.2** |

Table 2: Generalization performance of AFTER.

tively. The results in Table 2 demonstrate that AFTER still yields remarkable improvement under different image and question distributions. This indicates that AFTER can achieve general language bias mitigation of LVLMs rather than merely fitting a specific dataset, therefore exhibiting strong potential for generalization to open-world scenarios.

## 4.3 IN-DEPTH ANALYSIS

**Analysis of Factual-augmented Text**   We employ two strategies: serving as LVLM's input, and steering as trusted activation, to demonstrate the superiority of FAS-derived factual-augmented textual description $t^+$ over simple descriptions $t^s$ (*e.g.* COCO Caption (Chen et al., 2015)). Table 3 reveals that simple captions, lacking substantial factual information, perform even 6.7% worse than visual image $x$ as direct input, and offer marginal guidance in trusted editing. In contrast, our factual textual description encompasses extensive facts, leading to significantly fewer hallucinations than visual images. Furthermore, the visual-textual steering vector derived from FAS more effectively

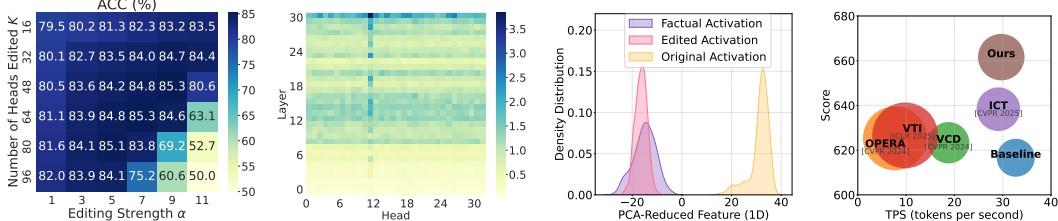

Figure 4: More analysis on LLaVA-v1.5. Subfigures are labeled sequentially as (a)-(d). (a) Ablation of number $K$ and strength $\alpha$. (b) Distribution of vector magnitudes. (c) Visualization of distinct activations yielded by the last layer. (d) Comparison of inference speed and mitigation results.

mitigates visual-textual disparity than those from simple captions, demonstrating superior guidance for reducing language bias.

Additionally, there is minimal performance variation between fact-augmented descriptions $t^+$ generated by LVLMs $\mathcal{F}$ with different parameters and architectures. This demonstrates that the $\mathcal{F}$ employed by FAS is solely utilized for integrating discrete facts into coherent textual ground truth, without distilling new knowledge from $\mathcal{F}$ that would influence the inference of the edited model.

| Input Semantics | Direct Input | | Steering Vector | |
|---|---|---|---|---|
| | ACC | F1 | ACC | F1 |
| Image $x$ | 79.2 | 80.9 | - | - |
| Simple Caption $t^s$ | 72.5 | 72.8 | 81.4 | 82.2 |
| $t^+$ GPT-4o (200B) | 93.4 | 93.4 | **85.3** | 84.4 |
| $t^+$ GPT-4o-mini (8B) | **93.9** | **93.7** | **85.3** | **84.5** |
| $t^+$ llava-v1.5 (7B) | 92.6 | 92.4 | 85.1 | 84.1 |

Table 3: Comparison of diverse inputs under two strategies. We analyze three variants of $\mathcal{F}$ with varying parameters and architectures for generating factual-augmented $t^+$.

**Analysis of Hyperparameter**  We analyze two hyperparameters that regulate the editing, *i.e.* the number of edited heads $K$ and editing strength $\alpha$. From Figure 4(a), we can observe that both the accuracy and F1 score exhibit an inverted U-shaped curve. The best accuracy (85.3%) is achieved at $K = 64$, $\alpha = 7$, while the highest F1 score (84.7%) appears at $K = 64$, $\alpha = 9$. These results demonstrate the effectiveness of editing with appropriately calibrated editing strength. The declines under excessive steering reveal a trade-off between truthfulness and helpfulness for editing methods  (Li et al., 2024a; Chen et al., 2025), providing us with intuitive guidance for editing.

**Analysis of Magnitude Distribution**  To investigate the impact of language bias within the LVLM architecture, we analyze the distribution of editing vector magnitudes across all layers and attention heads, as shown on Figure 4(b). The results reveal a notable increase in vector magnitudes in the middle layers (layers 9 to 17), which can be attributed to the progressive accumulation of visual information through self-attention  (Jiang et al., 2024c). Therefore, language bias significantly interferes with the perception of visual content, resulting in substantial visual-textual disparity. This effect accumulates across subsequent layers and ultimately propagates to the final layer, directly contributing to hallucinatory outputs. Moreover, we observe a particularly pronounced disparity at the 12th head, which may result from its heightened involvement in extracting visual object semantics.

**Visualization of Activations**  To qualitatively investigate the mechanism of AFTER, we visualize the distributions of the last layer's factual textual activations, along with original and post-edited activations via one-dimensional PCA projections in Figure 4(c). It is evident that the original visual activations exhibit significant divergence from the factual textual activation distribution, highlighting the initial visual-textual disparity that leads to hallucination. After applying adaptive factual-guided visual-textual editing, the visual activations shift notably towards the textual cluster, providing evidence that AFTER indeed offers effective guidance to steering visual activations towards factual textual semantics, achieving successful mitigation of language bias.

**Inference Computation**  We also compare inference speed and hallucination mitigation results on MME against other inference-time methods. Results in Figure 4(d) demonstrate that our AFTER

achieves the best mitigation performance while maintaining the fastest inference speed of 29.7 tokens per second. In addition, AFTER maintains moderate memory usage of 16.3 GB (expressed as the volume of spheres), facilitating practical deployment without demanding excessive resources.

## 5 CONCLUSION AND FUTURE WORK

In this paper, we propose AFTER, an effective activation editing approach that adaptively steers visual activation toward factual-augmented textual semantics for hallucination mitigation. Extensive experiments on typical hallucination benchmarks across three widely adopted LVLMs have confirmed that our AFTER achieves superior mitigation performance with minimal cost. It also exhibits strong generalizability and preserves general visual capabilities. A limitation of AFTER, as well as other activation editing methods, is its dependence on the accessible activations from open-source LLMs, which restricts its applicability to closed-source LLMs. Additionally, for tasks requiring substantial domain expertise, such as medical report analysis, AFTER necessitates supplementary domain-specific data to enhance LVLM's specialized visual perception and better mitigate language bias. In future work, we intend to extend AFTER to encompass a wider range of specialized domains.

## 6 ETHICS STATEMENT

We adhere to the ICLR Code of Ethics and confirm that the work presented in this paper complies with all applicable ethical standards. Our research does not involve human subjects, and all datasets used in the experiments are publicly available or obtained through ethical practices. We commit to responsible practices in the use of data and algorithms. No conflicts of interest or sponsorship issues have been identified.

## ACKNOWLEDGMENTS

This work was supported by the National Key Research and Development Program of China (NO. 2023YFC2506800), the Young Elite Scientists Sponsorship Program of the Beijing High Innovation Plan, and the Fundamental Research Funds for the Central Universities.

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

APPENDIX CONTENT

| Methods | Additional Training | Constructing Vectors/500 sample | Inference | Total |
|---------|---------|---------|---------|-------|
| HACL | 12.60 | - | **0.61** | 13.21 |
| VTI | - | 0.42 | 1.03 | 1.45 |
| ICT | 0.13 | 0.21 | 0.76 | 1.13 |
| Ours | **0.08** | **0.12** | 0.72 | **0.92** |

Table 4: Comparison of end-to-end computational cost (measured in GPU-hours) of POPE adversarial evaluation on LLaVA-v1.5 across various mitigation approaches, including training-based HACL, editing-based VIT and ICT. All experiments are conducted on 80G NVIDIA A100.

# A    More Experimental Analysis

## A.1    Analysis of Training Size

To explore AFTER's sensitivity to the training data, we vary the number of images used to construct vectors and train the offset estimator from 50 to 500. We also include experiments with fewer 5, 10, 20, and 30 training samples. As shown in Figure 5, our method consistently achieves strong performance across different scales, with accuracy rising from 79.2% to 85.3% and F1-score steadily improving from 83.6% to 84.5%. Despite the reduction in training data, we also sustain a relatively high performance level, outperforming the baseline by a large margin. Notably, even when using only 5 samples to construct factual-guided editing vectors and train QAO, our method achieves ACC and F1 scores of 81.0% and 82.1%, outperforming the baseline LLaVA-v1.5 by 1.8% and 1.2%. This is credited to the valuable assistance of augmented facts and query-specific editing, highlighting the practical advantage of AFTER.

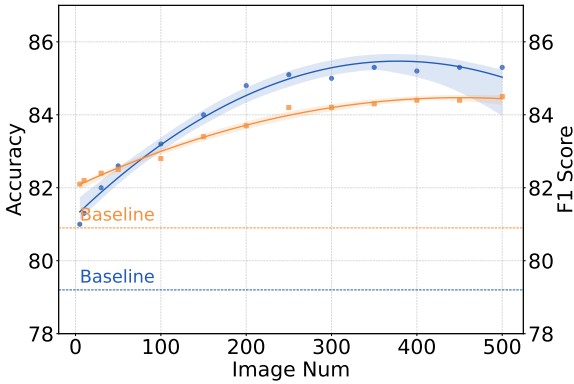

Figure 5: LLaVA-v1.5 performance on POPE across varying training data amounts.

## A.2    More Analysis of Efficiency

As detailed in Section 3.3, training the offset generators is highly efficient, being lightweight and requiring no fine-tuning of the LVLM. To further demonstrate resource efficiency, particularly in estimator training, we compare its resource consumption with that of HACL, a training-based fine-tuning method. The results in Table 4 demonstrate that on LLaVA-v1.5, our estimator training consumes only 0.08 GPU-hours, substantially less than HACL's 12.6 GPU-hours. Additionally, compared to other editing-based methods, we achieve the least computational cost across all process for activation editing. Notably, the estimator's training accounts for merely 8.7% of the overall computational cost, significantly undercutting the inference cost of the LVLM at test time. This highlights the lightweight and scalable nature of our pipeline, especially the minimal computational resources required for estimator training.

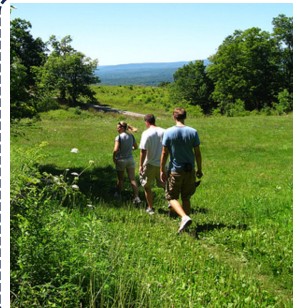

**Question:** Describe this image.

**Hallucinated Objects:** **Backpack**

**LLaVA-v1.5 Answer:** ...The field is surrounded by trees, providing a serene and natural setting for their walk. ***The people are carrying backpacks,*** which suggests they might be on a hike or a day trip. ...

**Ours Answer:** ...The field is lush and green, providing a beautiful backdrop for their walk. ***The people are walking in a straight line, with each person maintaining a comfortable distance from the others***. ...

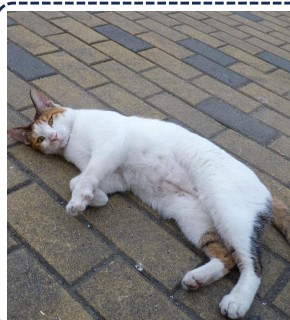

**Question:** Describe this image.

**Hallucinated Objects:** **Sun**

**LLaVA-v1.5 Answer:** The image features a white and orange cat lying on a brick walkway, possibly a sidewalk. The cat is stretched out and appears to be relaxed, ***enjoying the warmth of the sun***. ...

**Ours Answer:** The image features a white and orange cat lying on a brick floor, possibly a sidewalk. The cat appears to be relaxed and ***enjoying its time on the ground***. ...

Figure 6: Two case studies of LLaVA-v1.5 on AMBER.

## A.3 CASE STUDIS ON AMBER

In Figure 6, we present two representative case studies of LLaVA-v1.5 on the AMBER benchmark to intuitively demonstrate the effectiveness of AFTER in mitigating language bias. In the first case, LLaVA-v1.5 erroneously predicts that the people are carrying backpacks, likely influenced by the overall travel-related context of the image. In contrast, AFTER successfully avoids this misleading inference by neutralizing the "travel" prior embedded in the language model, and accurately describes the factual action of walking. In the second case, LLaVA-v1.5 hallucinates that the cat is enjoying the warmth of the sun, despite the image exhibiting a dim, overcast appearance. AFTER, being less susceptible to language priors, correctly avoids such hallucinated content and grounds its response in the actual visual cues.

## A.4 MORE ANALYSIS OF QAO

To further investigate the contribution of QAO to overall performance, we conducted an in-depth analysis comparing different offset supervision strategies for training QAO. The compared variants include: 1) Fixed strategy, using a constant vector added to the general steering vector $\mathbf{d}$. 2) Random strategy, replacing the QAO-predicted offset with a random vector sampled from a zero-mean Gaussian distribution. 3) Complete description $t^+$ guided strategy, constructing the ground-truth offset $o_i$ using the full description $t^+$ during QAO training. 4) Query-focused description $t^*$ guided strategy, constructing $o_i$ based on the query-focused description $t^*$, which constitutes our full QAO module. The comparative results are summarized in Table 5. We observe that fixed or random strategies fail to provide consistent improvements and can even degrade performance, indicating that factual-description guidance is essential for constructing meaningful, semantically aligned offset. Moreover, compared with complete description $t^+$ guided offset, the query-focused offset achieves the best performance on both POPE and AMBER, confirming that tailoring the offset to query-relevant semantics is crucial for maximizing the effectiveness of activation editing.

We further provide a set of visualizations to illustrate the role and effectiveness of the QAO mechanism. Specifically, we examine: (i) the t-SNE distribution of query-specific editing vectors obtained by adding QAO-generated offsets to the general steering vector $\mathbf{d}$. (ii) the PCA-reduced 1D feature distributions of LVLM's last-layer activations after edited with steering vector with and without QAO-generated offsets, and compare them with the disctribution of factual activation. These visualizations

| Methods | | POPE(ACC↑) | AMBER(Hal↓) |
|---|---|---|---|
| **w/o QAO** | | 83.8 | 22.3 |
| **w QAO** | Fixed | 81.6 | 28.9 |
| | Random | 83.2 | 23.6 |
| | Complete description $t^+$ guided | 84.9 | 20.9 |
| | Query-focused description $t^*$ guided | **85.7** | **20.5** |

Table 5: Comparison of different offset supervision strategies for training QAO on llava-v1.5.

| Different Diversity of $t^+$ | Direct Input | | Steering Vector | |
|---|---|---|---|---|
| | ACC | F1 | ACC | F1 |
| low-quality | 93.6 | **93.9** | **85.3** | 84.4 |
| high-quality | **93.9** | 93.7 | **85.3** | **84.5** |

Table 6: Comparison the editing performance of different quality and diversity of $t^+$ for llava-v1.5.

are presented in Figure 7. As shown in the right-hand t-SNE plot, the query-specific vectors produced by QAO form an approximately spherical Gaussian distribution around the general vector $\bar{\mathbf{d}}$. This pattern indicates that QAO successfully generates diverse offsets that vary meaningfully in both direction and magnitude. More importantly, the left two plots show that applying the QAO-generated query-specific offset systematically shifts the model's activations closer to the factual textual activation. This demonstrates that QAO adaptively guides the edited activations toward factually grounded regions of the activation space, thereby enhancing hallucination mitigation performance.

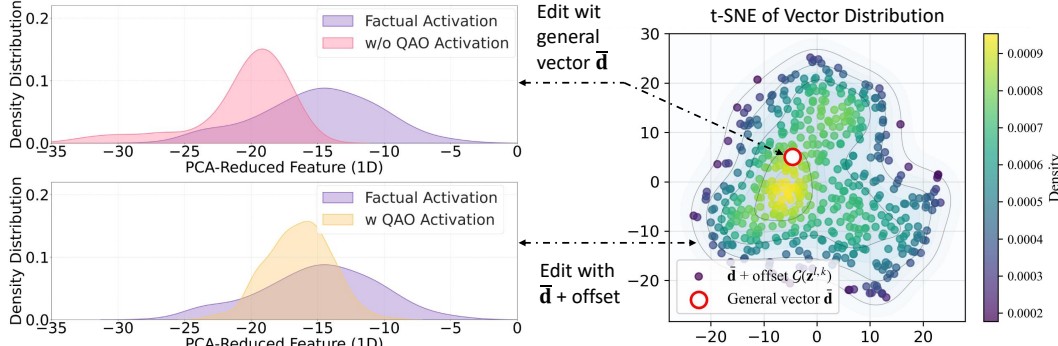

Figure 7: Visualization of QAO mechanism. **Left**: the PCA-reduced 1D feature distributions of LVLM's last-layer activations after edited with steering vector with and without QAO-generated offsets. **Right**: the t-SNE distribution of query-specific editing vectors obtained by adding QAO-generated offsets to the general steering vector $\bar{\mathbf{d}}$.

## A.5 MORE ANALYSIS ON DIVERSITY OF $t^+$

Diversity has a much smaller impact on performance. To validate these observations, we design a controlled experiment. Specifically, we prompt GPT-4o to remove all non-essential diversified information apart from GT from the original high-quality t+, producing low-diversity yet factually accurate descriptions. As shown in Table 6, when factuality is held constant, high-diversity and low-diversity t+ yield nearly identical results under both Direct Input and Steering Vector settings.

## A.6 PERFORMANCE ON MORE CHALLENGING BENCHMARK

we additionally conducted experiments on CRPE (Wang et al., 2024b) under the same experimental settings. The results are present in Table 7. The results show that AFTER achieves consistent improvements over the baseline across all evaluated dimensions, with notable 3.6% gains of Relation

task in the Object dimension. This indicates that our method can still accurately captures the presence of objects and their relationships in a more challenging context.

| Methods | Existence | Relation | | | |
|---|---|---|---|---|---|
| | | Subject | Predicate | Object | Overall |
| **Baseline** | 82.8 | 50.2 | 55.2 | 64.0 | 56.5 |
| **Ours** | **84.9** | **51.8** | **57.8** | **67.6** | **59.1** |

Table 7: Performance for LLaVA-v1.5 on more challenging CRPE benchmark.

## A.7    PERFORMANCE ON DATA-SCARCE SCENARIO

To further evaluate the applicability of our approach in domains where high-quality annotations are scarce or costly, we conduct additional experiments on the SLAKE dataset (Liu et al., 2021) using the medical LVLM LLaVA-med-v1.5 (Li et al., 2023a). SLAKE is a well-adopted medical VQA benchmark that provides COCO-like bounding-box annotations for organs and diseases, making it suitable for generating factual textual descriptions using AFTER. Specifically, we use the "Visual Misinterpretation" query set for SLAKE from MedHEval (Chang et al., 2025) to evaluate the hallucination mitigation performance of AFTER. Following the same experimental setup, we construct editing vectors using 5, 10, 30, and 50 samples, and the corresponding results are reported in Table 8. We can also observe a improvement of at least 1.0% in ACC when using only 5 samples, reinforcing the feasibility and effectiveness of our method in annotation-scarce domains.

| Methods | | ACC |
|---|---|---|
| **Baseline** | | 50.3 |
| **Ours** | 5 samples | 51.3 |
| | 10 samples | 51.6 |
| | 30 samples | 51.9 |
| | 50 samples | **52.0** |

Table 8: Hallucination mitigation performance for medical LVLM LLaVA-med-v1.5 on data-scarce medical analysis benchmark SLAKE.

## A.8    PERFORMANCE ON NOVEL CLASS

The steering direction learned by AFTER is class-agnostic and general, representing the comprehensive associations between textual and visual information within LVLMs. To further illustrate this, we evaluate the steering direction trained on COCO's 80 base classes using two subsets of GQA: (i) the subset corresponding to the same 80 base classes, and (ii) a subset of 47 novel classes sourced from YFCC100M (Thomee et al., 2016) (e.g.,rock, door, railroad) that do not appear in COCO. This setup simulates the experimental configuration for frequent versus rare classes. The results are listed in Table 9. The results demonstrate that AFTER achieves even higher ACC improvements on novel classes than on previously seen base classes. This can be attributed to the combination of lower baseline performance on novel classes and the inherently category-agnostic nature of our factual-guided steering vector.

| Classes | Methods | LLaVA-v1.5 | InstructBLIP | Shikra | Qwen2.5-vl-8B-Instruct |
|---|---|---|---|---|---|
| **Base** | **Baseline** | 89.5 | 88.0 | 84.6 | 87.9 |
| | **Ours** | **92.3** | **89.5** | **88.7** | **90.1** |
| **Novel** | **Baseline** | 66.7 | 71.9 | 74.4 | 80.0 |
| | **Ours** | **78.5** | **76.2** | **77.2** | **83.8** |

Table 9: Comparison of AFTER's performance on base classes and novel classes

| Models | Methods | POPE | | MME | | | | AMBER | | |
|---|---|---|---|---|---|---|---|---|---|---|
| | | ACC(↑) | F1(↑) | E(↑) | CT(↑) | P(↑) | CR(↑) | CH(↓) | Hal(↓) | Cr(↑) |
| LLaVA-v1.5 | Baseline | 80.1 | 82.3 | 180.0 | 158.3 | 123.3 | 155.0 | 6.9 | 31.6 | **48.9** |
| | Octopus | 83.2 | 84.0 | 180.0 | 153.3 | 128.7 | 155.0 | 5.2 | 24.6 | **48.9** |
| | VISTA | 83.9 | 84.1 | **195.0** | 161.7 | 116.7 | 155.0 | 5.1 | 24.4 | 48.4 |
| | Ours | **85.7** | **85.6** | **195.0** | 163.3 | 138.3 | 165.0 | **4.5** | 20.5 | 48.7 |
| Qwen2.5-vl-8B-Instruct | Baseline | 83.7 | 82.2 | 200.0 | 155.0 | 160.0 | 195.0 | 5.3 | 27.2 | 63.6 |
| | Ours | **87.4** | **87.0** | **200.0** | 160.0 | 165.0 | 195.0 | **3.6** | 24.0 | 63.7 |

Table 10: Comparison of AFTER with more methods on LLaVA-v1.5 and Qwen2.5-vl-8B-Instruct.

## A.9 PERFORMANCE ON MORE MODELS AND BASELINES

We additionally compared AFTER with the ICML 2025 activation editing method VISTA (Li et al., 2025) and the CVPR 2025 contrastive decoding method Octopus (Suo et al., 2025) on LLaVA-v1.5, and further evaluated AFTER on the 2025-released Qwen2.5-VL-8B-Instruct model (Bai et al., 2025b). The corresponding results are summarized in Table 10. As shown in the table, our method consistently surpasses the most recent activation-editing and contrastive decoding baselines, while delivering notable improvements on Qwen2.5-VL-8B-Instruct.

## A.10 ANALYSIS OF DIFFERENT EDITING LAYERS

Furthermore, we independently correct on the shallow, middle, and deep layers and observe that editing middle and deep layers consistently outperforms corrections applied to the shallow layers. Concretely, we use LLaVA-v1.5 as our test model and divide it into shallow (Layers 0–11), middle (Layers 12–23), and deep (Layers 24–31) segments. For each segment, we select the top-$K \in \{16, 32, 48, 64\}$ heads ranked by vector magnitude for editing, and set the editing strength to $\alpha = 7$. We evaluate POPE under the COCO adversarial setting, with ACC results reported in Table 11. It is observed that corrections on the middle and deep layers consistently outperform those on the shallow layers across all values of K. This is because the shallow layers primarily encode low-level perceptual features, where language-driven biases have not yet accumulated. This observation aligns with our analysis in Figure 4(b). Moreover, both the middle and deep layers exhibit substantial and progressively compounded cross-modal interactions and language-induced bias, leading to stronger and more stable mitigation effects of AFTER.

| Layers | $K = 16$ | $K = 32$ | $K = 48$ | $K = 64$ |
|---|---|---|---|---|
| Shallow | 81.5 | 82.0 | 82.7 | 82.3 |
| Middle | **82.8** | 83.7 | **85.0** | 84.9 |
| Deep | 82.7 | **84.0** | 84.8 | **85.1** |

Table 11: Comparison of the editing performance on different layers.

## B MORE DETAILS ON BENCHMARK EVALUATION

### B.1 POPE

POPE (Li et al., 2023b) evaluation benchmark is specialized to assess how effectively LVLMs identify object hallucinations. In this benchmark, LVLMs are tasked with determining whether a specified object is present in a given image. The ratio of queries targeting existent versus non-existent objects is balanced. POPE comprises three sampling settings—random, popular, and adversarial—which differ in their strategies for constructing negative samples. In the random setting, non-existent objects are sampled uniformly at random. The popular setting selects missing objects from a pool of frequently occurring categories, while the adversarial setting focuses on semantically co-occurring objects that are absent from the image. The benchmark draws data from three diverse sources: COCO (Lin et al., 2014), A-OKVQA (Schwenk et al., 2022), and GQA (Hudson & Manning, 2019). For each sampling setting, 500 images are sampled from each dataset, with a set of questions constructed per image, resulting in a total of 27,000 query-answer pairs derived from the development sets. Consistent

with previous work (Chen et al., 2025; Bai et al., 2025a; Liu et al., 2025), we report Accuracy and F1-score for comparison.

### B.2 MME

MME (Fu et al., 2023) benchmark is designed to comprehensively evaluate the general perception and cognition capability of LVLMs across 14 subtasks. Specifically, the perception capabilities involve coarse-grained existence, count, position, and color, which are used for hallucination evaluation by numerous studies (Leng et al., 2024; Chen et al., 2025; Zhuang et al., 2025), and fine-grained artwork(A), celebrity(C), landmark(L), OCR(O), posters(P), and scene(S). The short name corresponds with those in Figure 3. The cognition tasks include code reasoning(CdR), commonsense reasoning(CoR), numerical calculation(NC), and translation(T) to evaluate if LVLM can carry out further logical reasoning after perceiving the image. We adopt the official MME score as the comprehensive evaluation metric, providing a quantitative measure of the LVLM's performance across diverse tasks.

### B.3 AMBER

AMBER (Wang et al., 2023) is a meticulously designed benchmark, which introduces a low-cost, LLM-free pipeline to assess hallucinations across both generative and discriminative tasks, enabling comprehensive multi-dimensional analysis. In our experiments, we primarily focus on the generative subset for evaluation. AMBER specifically introduces four metrics for evaluating hallucinations on the generative task: CHAIR measures the frequency of hallucinatory objects appearing in the responses; Cover measures the object coverage of responses; Hal represents the proportion of responses with hallucinations and Cog assesses whether the hallucinations in MLLMs are similar to those in human cognition. We specifically employ CHAIR and Hal to evaluate hallucination degree, as well as Cover to assess the comprehensiveness.

## C More details on Implementation

### C.1 Detailed Construction of Color Fact

As described in Section 3.2, the color attribute facts are manually annotated based on pixel-level statistics within each object. Specifically, we first identify all pixel positions within the target region using a scanline fill algorithm applied to the object's segmentation polygon. For each pixel, we compute the Euclidean distance between its RGB value and those of the 17 standard base colors defined by CSS. The pixel is then assigned to the closest standard color. Finally, the color with the highest pixel count within the region is selected as the object's representative color fact.

### C.2 Detailed Construction of Shape Fact

To infer the semantic shape type of an object, we first extract its contour from the segmentation mask $S$ and approximate it using the Ramer–Douglas–Peucker algorithm to obtain a simplified polygonal representation. By examining key geometric properties of this polygon—such as the number of vertices, aspect ratio, internal angles, and edge-length uniformity—we classify the object into shape categories including triangle, rectangle, square, circle, ellipse, and irregular. For instance, a polygon with three vertices is identified as a triangle, while a four-vertex polygon with near-right angles and uniform sides is recognized as a square. Additionally, circularity and ellipticity metrics are computed based on area–perimeter relations and ellipse fitting, further aiding the classification of curved shapes. This geometric reasoning bridges low-level segmentation with high-level interpretable shape attributes.

### C.3 Detailed Construction of Relation Fact

To infer spatial relations between object pairs, we first compute the centroid of each object's bounding box and analyze the relative offset vector between them. By measuring the angle formed by this vector with respect to the horizontal axis, we discretize the 2D space into directional sectors—namely, top, bottom, left, right, and diagonals such as top-left or bottom-right. This angular quantization allows for interpretable directional labeling. Additionally, we assess the degree of overlap using

the intersection-over-union (IoU) between bounding boxes to identify occlusion or containment relationships. Together, these geometric cues enable a principled classification of object-to-object spatial relations into nine categories, including positional and overlapping types.

### C.4 Details of Training Estimator

The offset generator $\mathcal{G}$ is designed to capture distinct deviations from the general steering vector, thereby establishing query-specific visual-textual associations. Specifically, an offset generator is trained for each attention head; each consists of a simple multilayer perceptron (MLP) with two linear layers and a GELU activation layer. Training is conducted with a batch size of 16, for 10 epochs, using a learning rate of 1e-10, and optimized via Adam.

### C.5 Hyperparameter Setting

Following (Chen et al., 2025), we use hyperparameter tuning to determine editing strength $\alpha$ and edited heads num $K$ and ensure reproducibility. It is noticed that due to the uninterpretable nature of LVLMs, existing activation editing methods cannot specify the optimal editing strength or the appropriate head num without established theory, and therefore rely on empirical hyperparameter tuning. Hyperparameter tuning was performed using a grid search to explore the possible combinations of key parameters systematically. The hyperparameters under consideration included $\alpha$ and $K$. Specifically, the search space was defined as the Cartesian product: $\{1, 3, 5, 7, 9\} \times \{16, 32, 48, 64\}$, where the first discrete set denotes the selected values of $\alpha$, which range was chosen to strike a balance between improving model trustworthiness and maintaining overall performance, and the second is for $K$, which allows for sparse editing (total 1024 heads) for both effectiveness and efficiency.

In practice, AFTER can maintain stable performance on new datasets when initialized with the same prior hyperparameters. Only minor adjustments are typically required to reach optimal performance.

## D Examples of Factual Textual Description

In this section, we present two examples illustrating the generation process and results of both general and query-specific factual textual descriptions in Figure 8 and 9.

## E Full Experimental Results

We present the full experimental results of POPE on COCO, A-OKVQA and GQA dataset in Table 12, Table 13 and Table 14 respectively, which constitute the averaged scores presented in Table 1.

## F More Details on Prompts

The prompt $\mathrm{I}_{\mathrm{fst}}$ used to textualize all the facts into a comprehensive and factual description $t^+$ based on the perception of the visual content is as follows:

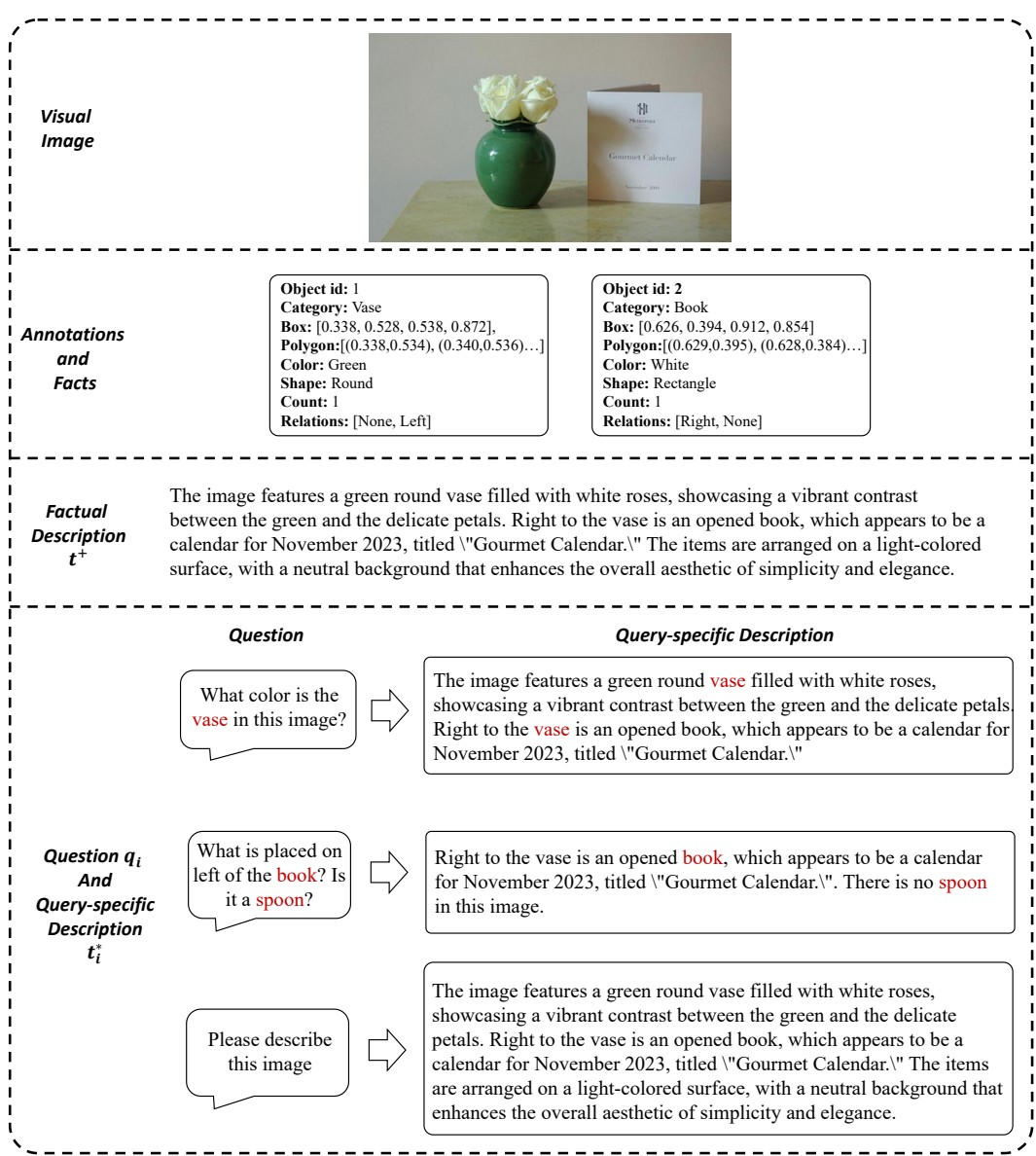

Figure 8: An example of the numerous facts extracted from annotations, generated factual description $t^+$, questions, and query-specific descriptions $t_i^*$.

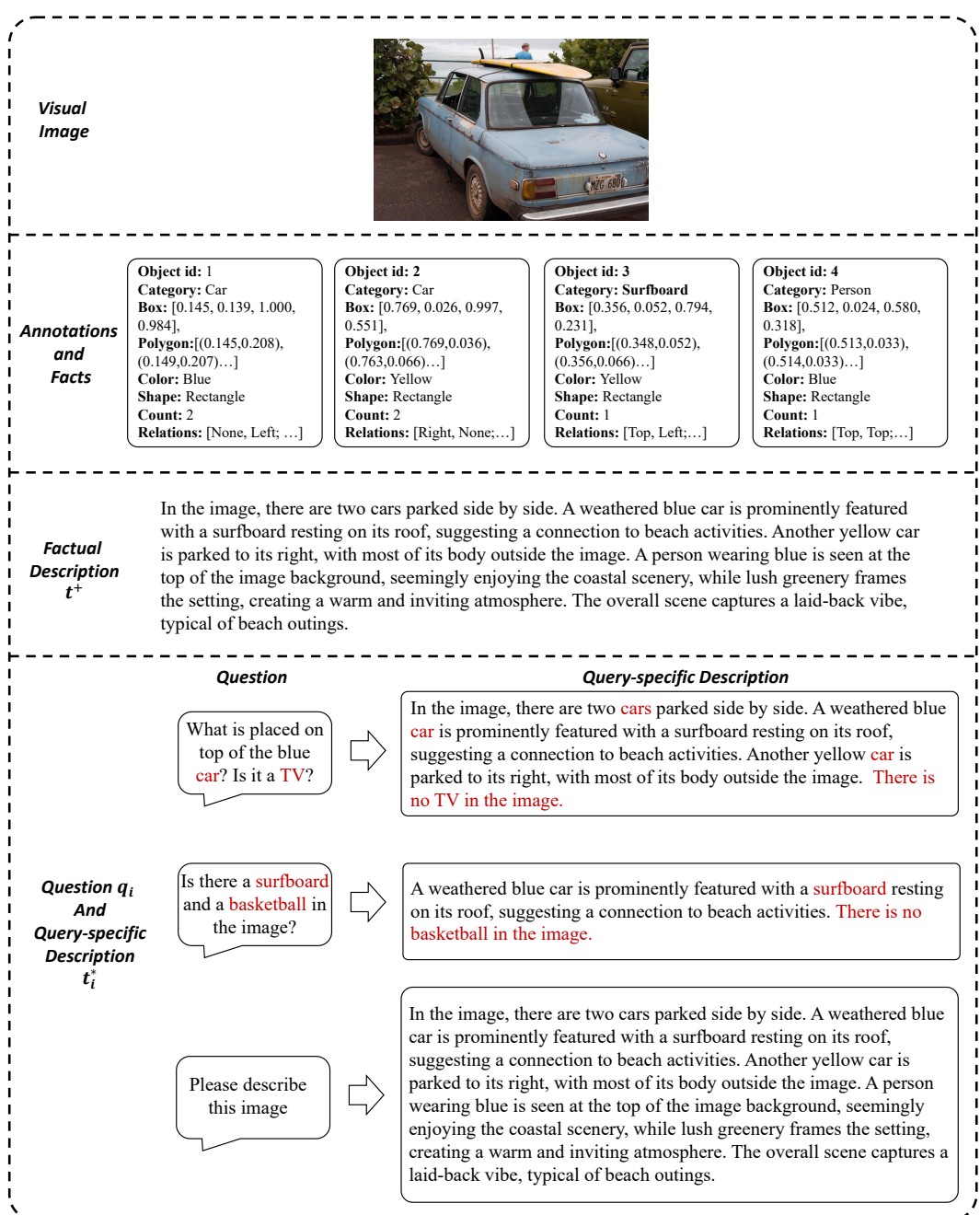

Figure 9: Another example of the numerous facts extracted from annotations, generated factual description $t^+$, questions, and query-specific descriptions $t_i^*$.

| Dataset | Settings | Methods | LLaVA-v1.5 ACC | F1 | InstructBLIP ACC | F1 | Shikra ACC | F1 |
|---------|----------|---------|------|------|------|------|------|------|
| COCO | Random | Baseline | 89.0 | 88.9 | 89.4 | 89.0 | 81.3 | 82.0 |
| | | HACL | 89.5 | 88.0 | - | - | 82.1 | 82.5 |
| | | VCD | 88.8 | 88.4 | 89.8 | 89.5 | 82.1 | 82.5 |
| | | OPERA | 88.9 | 88.9 | 89.9 | 90.0 | 82.9 | 82.9 |
| | | VTI | 89.8 | 89.3 | 90.8 | 90.0 | 83.2 | 83.2 |
| | | ICT | 89.6 | 89.5 | 90.8 | 90.2 | 83.8 | 83.7 |
| | | Ours *w/o* QAO | 89.7 | 89.3 | 90.5 | 90.1 | 84.0 | 83.7 |
| | | **Ours** | **90.1** | **89.9** | 90.6 | **90.2** | **85.5** | **84.5** |
| | Popular | Baseline | 85.6 | 85.9 | 82.4 | 83.0 | 81.2 | 81.9 |
| | | HACL | 85.9 | 85.3 | - | - | 81.4 | 82.0 |
| | | VCD | 85.7 | 85.7 | 83.0 | 83.3 | 82.2 | 82.4 |
| | | OPERA | 86.0 | 86.4 | 83.7 | 83.9 | 82.3 | 82.5 |
| | | VTI | 86.3 | 86.4 | 83.6 | 83.8 | 82.5 | 82.6 |
| | | ICT | 86.6 | 86.5 | 83.6 | 83.8 | 83.4 | 83.5 |
| | | Ours *w/o* QAO | 87.3 | 87.4 | 83.8 | 84.0 | 83.5 | 83.4 |
| | | **Ours** | **87.7** | **87.5** | **84.3** | **84.4** | **84.1** | **83.7** |
| | Adversarial | Baseline | 79.2 | 80.9 | 78.3 | 79.8 | 77.0 | 78.7 |
| | | HACL | 82.0 | 81.6 | - | - | 77.8 | 79.1 |
| | | VCD | 81.1 | 81.9 | 80.0 | 80.3 | 78.2 | 79.2 |
| | | OPERA | 82.1 | 82.3 | 80.9 | 81.0 | 78.5 | 79.0 |
| | | VTI | 81.6 | 82.1 | 81.1 | 81.3 | 78.6 | 79.3 |
| | | ICT | 83.0 | 83.1 | 81.3 | 81.6 | 78.9 | 79.8 |
| | | Ours *w/o* QAO | 83.7 | 83.4 | 82.0 | 82.3 | 79.5 | 80.0 |
| | | **Ours** | **85.3** | **84.5** | **82.5** | **82.4** | **80.8** | **80.9** |

Table 12: Full POPE results on COCO dataset across random, popular, and adversarial settings.

| Dataset | Settings | Methods | LLaVA-v1.5 ACC | F1 | InstructBLIP ACC | F1 | Shikra ACC | F1 |
|---------|----------|---------|------|------|------|------|------|------|
| A-OKVQA | Random | Baseline | 86.3 | 87.4 | 88.1 | 88.6 | 80.9 | 82.1 |
| | | HACL | 87.0 | 86.3 | - | - | 81.1 | 82.2 |
| | | VCD | 86.9 | 86.5 | 88.6 | 88.8 | 82.8 | 83.4 |
| | | OPERA | 87.6 | 88.0 | 89.1 | 89.2 | 82.8 | 83.3 |
| | | VTI | 86.9 | 87.8 | 89.0 | 88.9 | 83.6 | 83.9 |
| | | ICT | 87.6 | 88.1 | 88.9 | 88.9 | 83.5 | 83.7 |
| | | Ours w/o offset | 87.7 | 88.3 | 89.8 | 89.8 | 84.1 | 84.1 |
| | | **Ours** | **89.5** | **89.1** | **89.9** | **90.0** | **85.1** | **84.8** |
| | Popular | Baseline | 78.9 | 81.8 | 79.5 | 81.1 | 80.7 | 81.9 |
| | | HACL | 82.6 | 82.1 | - | - | 81.1 | 82.1 |
| | | VCD | 81.9 | 82.8 | 82.3 | 82.4 | 81.8 | 82.5 |
| | | OPERA | 82.3 | 82.9 | 82.3 | 82.6 | 81.4 | 82.3 |
| | | VTI | 82.3 | 82.8 | 82.0 | 82.2 | 81.7 | 82.1 |
| | | ICT | 83.2 | 83.1 | 82.3 | 82.5 | 82.0 | 82.5 |
| | | Ours w/o offset | 82.0 | 83.4 | 82.9 | 83.9 | 81.9 | 82.2 |
| | | **Ours** | **85.4** | **85.2** | **84.1** | **84.7** | **83.1** | **83.1** |
| | Adversarial | Baseline | 71.2 | 74.9 | 71.3 | 75.3 | 73.8 | 76.0 |
| | | HACL | 77.3 | 76.3 | - | - | 73.9 | 76.9 |
| | | VCD | 76.1 | 76.0 | 74.2 | 76.0 | 74.6 | 77.3 |
| | | OPERA | 76.7 | 76.6 | 74.0 | 74.3 | 74.4 | 77.2 |
| | | VTI | 75.1 | 75.7 | 74.3 | 75.6 | 75.1 | 77.5 |
| | | ICT | 76.6 | 76.5 | 74.6 | 75.2 | 75.0 | 77.6 |
| | | Ours w/o offset | 77.1 | 78.2 | 74.9 | 78.1 | 75.3 | 77.0 |
| | | **Ours** | **79.1** | **80.1** | **76.2** | **78.9** | **76.9** | **78.2** |

Table 13: Full POPE results on A-OKVQA across random, popular, and adversarial settings.

| Dataset | Settings | Methods | LLaVA-v1.5 ACC | LLaVA-v1.5 F1 | InstructBLIP ACC | InstructBLIP F1 | Shikra ACC | Shikra F1 |
|---|---|---|---|---|---|---|---|---|
| GQA | Random | Baseline | 86.0 | 87.3 | 85.9 | 86.3 | 80.2 | 81.4 |
| | | HACL | 87.9 | 86.9 | - | - | 81.5 | 82.2 |
| | | VCD | 87.0 | 86.7 | 86.3 | 86.8 | 81.4 | 82.3 |
| | | OPERA | 88.1 | 88.3 | 87.0 | 86.8 | 81.7 | 82.4 |
| | | VTI | 87.9 | 88.0 | 87.5 | 87.2 | 82.4 | 82.9 |
| | | ICT | 88.3 | 88.3 | 87.6 | 87.6 | 82.8 | 83.1 |
| | | Ours w/o offset | 88.9 | 89.3 | 87.7 | 87.7 | 83.5 | 83.6 |
| | | **Ours** | **90.0** | **90.1** | **87.8** | **87.9** | **85.1** | **84.7** |
| | Popular | Baseline | 73.7 | 78.6 | 76.4 | 79.2 | 80.1 | 81.3 |
| | | HACL | 81.4 | 80.9 | - | - | 80.6 | 80.8 |
| | | VCD | 79.1 | 79.6 | 78.1 | 79.1 | 81.1 | 81.9 |
| | | OPERA | 81.0 | 81.4 | 78.2 | 79.5 | 81.5 | 82.0 |
| | | VTI | 81.3 | 81.3 | 78.7 | 80.1 | 81.6 | 82.0 |
| | | ICT | 81.1 | 80.9 | 79.1 | 79.6 | 82.1 | 82.2 |
| | | Ours w/o offset | 80.6 | 81.3 | 79.5 | 80.5 | 81.3 | 81.8 |
| | | **Ours** | **83.5** | **83.4** | **80.2** | **81.2** | **83.1** | **83.2** |
| | Adversarial | Baseline | 71.1 | 75.1 | 71.5 | 76.0 | 74.9 | 77.5 |
| | | HACL | 78.1 | 77.7 | - | - | 75.8 | 78.0 |
| | | VCD | 76.4 | 77.1 | 73.3 | 75.1 | 76.5 | 78.4 |
| | | OPERA | 76.6 | 77.0 | 74.9 | 76.0 | 76.3 | 78.3 |
| | | VTI | 77.3 | 77.6 | 74.8 | 76.6 | 76.8 | 78.5 |
| | | ICT | 77.1 | 77.0 | 75.1 | 76.5 | 76.9 | 78.5 |
| | | Ours w/o offset | 77.0 | 78.8 | 75.4 | 77.5 | 77.0 | 78.9 |
| | | **Ours** | **80.3** | **80.8** | **76.2** | **78.2** | **78.7** | **79.5** |

Table 14: Full POPE results on GQA across random, popular, and adversarial settings.

> Your task is to generate comprehensive and factual description of the given image based on the factual information in a single paragraph. Specifically, the first step is to interpret the content of the given image. The second step is to generate a complete and accurate image description by integrating the factual information of all provided objects. Each object's factual information includes its category, position, color, shape, count, and relationships with other objects. In the generated description, ensure that all provided factual information is included as comprehensively as possible.
>
> Factual Information:
> {
> Object 1:[
>   Category: [Category]
>   Location: [Location]
>   Color: [Color]
>   Shape: [Shape]
>   Count: [Count]
>   Relations: [Relation 1, Relation 2, ...]
> ],
> ... }
>
> Output Description:

The prompt $I_{qst}$ used to generate object-related textual description $t_{i,j}^+$ from the whole description through GPT-4 as follows:

> Your task is to extract and return the related description of the given category. For each category, retain only the parts of the original description that are relevant to the category. Finally, output the object-related textual descriptions in the same order as the input objects, separated by semicolons.
>
> Original description: [Paragraph]
> Objects: [Object 1, Object 2, ...]
> Object-related description:

## G  BROADER IMPACT

The proposed AFTER framework offers substantial benefits in enhancing the trustworthiness of Large Vision-Language Models (LVLMs). By effectively mitigating object hallucinations—particularly those stemming from language priors—AFTER enables models to generate outputs that more faithfully reflect the underlying visual content. This is crucial for fostering user confidence in LVLM-driven systems across a wide range of applications, such as image captioning, human-AI interaction, and assistive technologies. AFTER's factual-guided visual-textual editing helps the model better align its internal representations with ground-truth facts, reducing discrepancies that often confuse or mislead users. Additionally, its query-adaptive mechanism ensures context-aware correction rather than static interventions, leading to fine-grained and semantically consistent outputs. One notable advantage is that AFTER achieves these improvements with minimal computational cost and without requiring model retraining, making it highly deployable across diverse platforms and resource environments. Its plug-and-play nature allows existing systems to adopt it easily, contributing to broader accessibility and general improvement in the factual grounding of LVLMs. This positions AFTER as a practical and efficient step toward building more trustworthy multimodal AI systems.

Although the overall risk of misuse is limited, we acknowledge that factual-guided editing may inherit biases or omissions from the underlying factual sources. For example, if the facts used in steering vectors are incomplete or skewed, the resulting edits might inadvertently reinforce those limitations. However, these concerns can be effectively addressed by ensuring that factual data used in activation steering is diverse, high-quality, and periodically updated. Furthermore, we recommend applying domain-specific factual augmentation and validation protocols when deploying AFTER in specialized areas, such as biomedical or legal contexts. These precautions help ensure that the editing process remains aligned with accurate, context-appropriate semantics. With proper oversight, AFTER can be a robust and broadly applicable tool for enhancing the integrity of LVLM outputs.

## H  LLM USAGE

This paper made limited use of ChatGPT solely for language polishing and grammar refinement. No part of the conceptual development, methodology, analysis, or experimental results was generated by large language models. Since our research focuses on activation editing of LVLMs, LLMs appear as the experimental subjects in our study. Beyond this role as objects of investigation and minor linguistic editing, LLMs were not used for any other purpose in this work.

