# OpenReview forum: "AFTER: Mitigating the Object Hallucination of LVLM via Adaptive Factual-Guided Activation Editing"
_ICLR.cc/2026/Conference — ICLR 2026 Poster_

### Official Review · Reviewer_n12p · 2025-10-30

**Soundness:** 3
**Presentation:** 3
**Contribution:** 3
**Rating:** 4
**Confidence:** 4

**Summary:**

This paper proposes an activation editing method AFTER to mitigate hallucination in vision-language models. It compromises two methods: FAS to provide factual and global guidance for activation editing, and QAO to provide query-specific guidance. Through training a lightweight MLP, they obtain the offset activation vectors, which is added to the original activations for correction. Results show that AFTER outperforms baselines and previous methods.

**Strengths:**

1. The writing is mostly clear and easy to follow.

2. The method seems effective and substantially outperforms baseline.

**Weaknesses:**

1. In FAS, how do you construct an $n$-question set for each textual description while ensuring that each of $n$ questions are indeed effective? Besides, what is the value of $n$ used in your experiments?

2. What model is used to generate $t^{+}$? Also, despite $t^{+}$ is synthesized by a set of facts, hallucination can still persist in outputs of any LLMs, so validation of  $t^{+}$ is needed.

3. Models and baselines compared are outdated (all in 2024) considering this is a submission in late 2025.

4. The notation is quite confusing for  $t_{i,j}^{+}$ at the end of line 274. Correct me if I’m wrong: $i$ should be from 1 to $n$, denoting the $n$ questions associated with each sample; $j$ should be from 1 to $m$, which is the number of objects mentioned in the question $q_{i}$, not from 1 to $n$.

5. In line 277, $z_{i}^{\*}$ is obtained from $t_{i}^{*}$, which is a **set** of activations, but $z_{i}$ is a single activation. How is the subtraction between them possible?

6. How does the selection of training sets impact the trained MLP for offset prediction? Since all of the four benchmarks (POPE, MME, AMBER, GQA) contain images from MSCOCO, it’s necessary to use a different set of images for training or benchmarks with different image sources to further prove the generalization of your method.

**Questions:**

1. Are you correcting activations for all layers? Since the shallow layers are more for perception and deeper layers for cognition, it’s more interesting to see the correction effect on different layers.

2. The models used to generate training data are not at all mentioned.

---

> ### Author Response · Authors · 2025-11-20
> **Response to Reviewer n12p (1/4)**
>
> We are truly thankful for your comprehensive review and the appreciation of our effectiveness.
>
> > ***Weakness 1:** In FAS, how do you construct an n-question set for each textual description while ensuring that each of n questions are indeed effective? Besides, what is the value of n used in your experiments?*
>
> Thanks for your insightful comments. The question construction in FAS **strictly follows the number and protocols of the classic benchmarks POPE[1] and AMBER[2]**, ensuring that the generated questions and answers comprehensively cover open-ended and generative questions, objects that are present or absent in the GT, as well as descriptions of single and multiple objects.
>
> Specifically, for each textual description, we generate **n=7** questions: 6 are discriminative questions constructed following the POPE protocol, and 1 is a generative question modeled after AMBER.
>
> - The 6 POPE-based questions are constructed using the template:*“Is there a/an <object> in the image?”*, with 3 objects selected from GT categories present in the image and 3 objects selected from frequently co-occurring categories that are absent from the image (i.e. adversarial sampling strategy).
> - The one AMBER-based question mirrors the corresponding AMBER dataset prompt,*“Describe this image”*, where the answer includes descriptions of all objects present in the image.
>
> [1] Li Y, Du Y, Zhou K, et al. Evaluating Object Hallucination in Large Vision-Language Models[C]//Proceedings of the 2023 Conference on Empirical Methods in Natural Language Processing. 2023: 292-305.
>
> [2] Wang J, Wang Y, Xu G, et al. Amber: An llm-free multi-dimensional benchmark for mllms hallucination evaluation[J]. arXiv preprint arXiv:2311.07397, 2023.
>
> > ***Weakness 2:** What model is used to generate t+? Also, despite t+ is synthesized by a set of facts, hallucination can still persist in outputs of any LLMs, so validation of t+ is needed.*
>
> Thanks for your meaningful comment.**In our experiments, we use three different models GPT-4o (200B), GPT-4o-mini (8B), and LLaVA-v1.5 (7B) as the generation model $F$**. More importantly, **AFTER explicitly constrains the generation process to avoid losing or distorting ground-truth information**, allowing the model $F$ to elaborate only within this boundary and thereby **preventing it from injecting its own hallucinations into t+**. We have included this interpretation in Appendix C.6 of the revised paper.
>
> **Both quantitative evaluation and manual inspection confirm that all generated texts are of high quality and exhibit no hallucinations.** To further substantiate this claim, we restate the experiments provided in Table 3, Section 4.3. below:
>
> |||||||
> |-|-|-|-|-|-|
> |Input Semantics||Direct Input||Steering Vector||
> |||ACC|F1|ACC|F1|
> |Simple Caption t^s||72.5|72.8|81.4|82.2|
> |t+  |GPT-4o(200B) |93.4|93.4|85.3|84.4|
> ||GPT-4o-mini(8B) |93.9|93.7|85.3|84.5|
> ||llava-v1.5(7B)|92.6|92.4|85.1|84.1|
>
>
> As shown in the table, the factual descriptions t+ generated by models of different sizes and architectures remain consistently high in quality. All generated t+ achieve direct-input accuracies above 92.6%, exceeding those of simple COCO captions t^s by more than 20%. In this experiment, the direct-input accuracies of t^+ **do not reach 100% primarily because the inference model(LLaVA-v1.5) has limited capability** in extracting question-relevant information from lengthy factual descriptions t^+, occasionally leading to incorrect answers.
>
> **Therefore, further validation of t+ offers limited practical value**. Naturally, for highly safety-critical domains such as medical analysis—where the correctness of GT must be rigorously guaranteed—human verification can be readily incorporated to ensure reliability.

---

> ### Author Response · Authors · 2025-11-20
> **Response to Reviewer n12p (2/4)**
>
> > ***Weakness 3:** Models and baselines compared are outdated (all in 2024) considering this is a submission in late 2025.*
>
> Thanks for your valuable comments. **Our compared baselines already include the most recent methods published in 2025, with remaining baselines and model selections representing the most classical and influential works in the field**:
>
> - **Activation editing baselines:** Our two core baseline methods, VTI and ICT, were published at **ICLR 2025[1]** and **CVPR 2025 [2]**, respectively, and were the most advanced activation editing approaches available at the time of submission. We have updated the citations of these methods to reflect their corresponding conference publications.
> - **Contrastive decoding baselines:** VCD [3] and OPERA [4] are among the most established contrastive decoding methods, collectively accumulating more than **786 citations according to Google Scholar**.
> - **Training-based baseline:** HACL [5] mitigates hallucinations through cross-modal alignment between visual and textual representations, making it conceptually related to our approach.
> - **Model choices:** LLaVA-v1.5, InstructBLIP, and Shikra are three of the most widely used LVLMs in hallucination research. To quantify this, we searched Google Scholar using the keywords *“hallucination mitigation, LVLM”* and inspected the top 50 relevant hallucination-mitigation papers from 2024 onward. Statistics show that LLaVA-v1.5 appeared in 49 papers **(Rank #1)**, InstructBLIP in 19 papers **(Rank #2)**, and Shikra in 11 papers **(Rank #5)**.
>
> Following your suggestion, we additionally compared AFTER with the **ICML 2025** activation editing method VISTA[6] and the **CVPR 2025** contrastive decoding method Octopus[7] on LLaVA-v1.5, and further evaluated AFTER on the 2025-released **Qwen2.5-VL-8B-Instruct** model [8]. The corresponding results are summarized in the table below.
> *(Note: Due to time constraints, we evaluated only the first 1,000 samples for each of the 9 POPE settings on Qwen2.5-VL-8B-Instruct.)*
>
> ||||||||||||
> |-|-|-|-|-|-|-|-|-|-|-|
> |Model  |Methods  |POPE||MME||||AMBER|||
> |||ACC(↑)|F1(↑)|E(↑)|CT(↑)|P(↑)|CR(↑)|CH(↓)|Hal(↓)|CH(上)|
> |LLaVA-v1.5  |Baseline|80.1|82.3 |180.0 |158.3 |123.3 |155.0|6.9 |31.6|**48.9**|
> ||Octopus |83.2|84.0|180.0|153.3|128.7|155.0|5.2|24.6|**48.9**|
> ||VISTA|83.9|84.1|**195.0**|161.7|116.7|155.0|5.1|24.4|48.4|
> ||Ours|**85.7**|**85.6**|**195.0**|**163.3**|**138.3**|**165.0**|**4.5**|**20.5**|48.7|
> |Qwen2.5-vl-8B-Instruct|Baseline|83.7|82.2|**200.0**|155.0|160.0|**195.0**|5.3|27.2|63.6|
> ||Ours|**87.4**|**87**|**200.0**|**160.0**|**165.0**|**195.0**|**3.6**|**24**|**63.7**|
>
>
> As shown in the table, our method consistently surpasses the most recent activation-editing and contrastive decoding baselines, while delivering notable improvements on Qwen2.5-VL-8B-Instruct. These additional results have now been included in the Appendix A.11.
>
>
>
> [1] Liu S, Ye H, Zou J. Reducing hallucinations in large vision-language models via latent space steering[C]//The Thirteenth International Conference on Learning Representations. 2025.
>
> [2]Chen J, Zhang T, Huang S, et al. Ict: Image-object cross-level trusted intervention for mitigating object hallucination in large vision-language models[C]//Proceedings of the Computer Vision and Pattern Recognition Conference. 2025: 4209-4221.
>
> [3]Leng S, Zhang H, Chen G, et al. Mitigating object hallucinations in large vision-language models through visual contrastive decoding[C]//Proceedings of the IEEE/CVF Conference on Computer Vision and Pattern Recognition. 2024: 13872-13882.
>
> [4]Huang Q, Dong X, Zhang P, et al. Opera: Alleviating hallucination in multi-modal large language models via over-trust penalty and retrospection-allocation[C]//Proceedings of the IEEE/CVF Conference on Computer Vision and Pattern Recognition. 2024: 13418-13427.
>
> [5]Jiang C, Xu H, Dong M, et al. Hallucination augmented contrastive learning for multimodal large language model[C]//Proceedings of the IEEE/CVF Conference on Computer Vision and Pattern Recognition. 2024: 27036-27046.
>
> [6]Li Z, Shi H, Gao Y, et al. The Hidden Life of Tokens: Reducing Hallucination of Large Vision-Language Models Via Visual Information Steering[C]//Forty-second International Conference on Machine Learning.
>
> [7] Suo W, Zhang L, Sun M, et al. Octopus: Alleviating hallucination via dynamic contrastive decoding[C]//Proceedings of the Computer Vision and Pattern Recognition Conference. 2025: 29904-29914.
>
> [8]Bai S, Chen K, Liu X, et al. Qwen2. 5-vl technical report[J]. arXiv preprint arXiv:2502.13923, 2025.

---

> ### Author Response · Authors · 2025-11-20
> **Response to Reviewer n12p (3/4)**
>
> > ***Weakness 4:** The notation is quite confusing for tij+ at the end of line 274. Correct me if I’m wrong: i should be from 1 to n, denoting the n questions associated with each sample; j should be from 1 to m, which is the number of objects mentioned in the question qi, not from 1 to n.*
>
> Thank you for pointing it out. The equation at the end of line 274 should indeed be corrected to $t_i^*=[t_{i,j}^+]_{j=1}^m$, where the index j should range from 1 to m. We have revised this accordingly in the paper and conducted a thorough consistency check to ensure that all related notation is correct throughout the manuscript.
>
> > ***Weakness 5:** In line 277, z_i∗ is obtained from t∗i, which is a set of activations, but z_i is a single activation. How is the subtraction between them possible?*
>
> Thanks for your valuable comments. We clarify that  $t_i^*=[t_{i,j}^+]^m_{j=1}$ denotes the concatenation of m textual segments $t_{i,j}$ into a single new textual description， rather than a set of activations. The bracket notation [ ] here specifically represents a text-concatenation operation. We have elaborated on this point in the revised manuscript to ensure clarity.
>
> > ***Weakness 6:** How does the selection of training sets impact the trained MLP for offset prediction? Since all of the four benchmarks (POPE, MME, AMBER, GQA) contain images from MSCOCO, it’s necessary to use a different set of images for training or benchmarks with different image sources to further prove the generalization of your method?*
>
> Thank you for providing such constructive feedback. **Our trained MLP (i.e., the QAO module) for offset prediction demonstrates stability under different selections of training sets.** This is because QAO learns the intrinsic correlations of language bias manifested in the activation space after textual-visual interactions within the LVLM. The impact of image domain shifts can be reduced since the embedded biases are primarily governed by the LVLM.
>
> **Our further COCO→BLINK generalization experiments, where the benchmark BLINK has different image sources with COCO, demonstrate that AFTER still achieves notable generalization performance to truly out-of-domain scenarios.**  Specifically, we evaluate cross-domain generalization with Qwen2.5-vl-8B-Instruct by applying editing vectors derived from COCO to the out-of-domain BLINK dataset. The experiments are shown in Appendix A.8 of the revised paper, with the results presented in the following table:
>
> |||||||||||||||||
> |-|-|-|-|-|-|-|-|-|-|-|-|-|-|-|-|
> |Methods|**Averaged**|OL|RR  |C  |RD|VC|AS|FC|IT|MR |SC|VS|FD|J|SR|
> |Baseline|54.5|51.6|38.8|69.2|64.5|54.1|**63.3**|**23.1**|26.0|**46.6**|34.5|88.9|46.2|**66.7**|**89.5**|
> |Ours|**55.7**|**56.9**|**40.3**|**73.6**|**66.2**|**54.6**|63.1|22.9|**27.2**|46.5|**35.9**|**89.0**|**47.6**|66.3|**89.5**|
>
>
> The results show that, relative to the baseline, our method achieves an average improvement of 1.2%, with particularly notable gains on tasks such as Object Localization and Counting. These findings demonstrate that our approach retains strong generalization ability even when applied to out-of-domain image sources. It is worth noting that the generalization performance of AFTER on **multi-image tasks** (e.g., Functional Correspondence(FC):–0.2%, Visual Correspondence(VC):+0.5%, Multi-view Reasoning(MR):–0.1%, among 8 tasks) is notably weaker than on **single-image tasks** (including **Object Localization(OL):+5.3%, Counting(C):+4.4%, Relative Depth(RD):+2.3%**, among 6 tasks). We conjecture that this may stem from the substantial discrepancy between multi-image and single-image input paradigms.
>
> [1]Chen J, Zhang T, Huang S, et al. Ict: Image-object cross-level trusted intervention for mitigating object hallucination in large vision-language models[C]//Proceedings of the Computer Vision and Pattern Recognition Conference. 2025: 4209-4221.
>
> [2]Hudson D A, Manning C D. Gqa: A new dataset for real-world visual reasoning and compositional question answering[C]//Proceedings of the IEEE/CVF conference on computer vision and pattern recognition. 2019: 6700-6709.
>
> [3]Thomee B, Shamma D A, Friedland G, et al. Yfcc100m: The new data in multimedia research[J]. Communications of the ACM, 2016, 59(2): 64-73.

---

> ### Author Response · Authors · 2025-11-20
> **Response to Reviewer n12p (4/4)**
>
> > ***Question 1:** Are you correcting activations for all layers? Since the shallow layers are more for perception and deeper layers for cognition, it’s more interesting to see the correction effect on different layers?*
>
> Thank you for your feedback. **As stated in lines 290–291, we apply correction only to the top-K attention heads most strongly influenced by language bias**, where the degree of bias is quantified by the vector magnitude. A larger magnitude indicates a stronger bias effect. This selective editing strategy preserves computational efficiency while maintaining hallucination suppression performance and minimizing unintended interference with the model's normal behavior.
>
> Furthermore, we independently correct on the shallow, middle, and deep layers and **observe that editing middle and deep layers consistently outperforms corrections applied to the shallow layers.** Concretely, we use LLaVA-v1.5 as our test model and divide it into shallow (Layers 0–11), middle (Layers 12–23), and deep (Layers 24–31) segments. For each segment, we select the top-𝐾 \in {16,32,48,64} heads ranked by vector magnitude for editing, and set the editing strength to α=7. We evaluate POPE under the COCO adversarial setting, with ACC results reported in Appendix A.12 of the revised paper.
>
> ||||||
> |-|-|-|-|-|
> ||K=16|K=32|K=48|K=64|
> |Shallow Layer|81.5|82.0|82.7|82.3|
> |Middle Layer|**82.8**|83.7|**85.0**|84.9|
> |Deep Layer|82.7|**84.0**|84.8|**85.1**|
>
>
> It is observed that corrections on the middle and deep layers consistently outperform those on the shallow layers across all values of K. This is because the shallow layers primarily encode low-level perceptual features, where language-driven biases have not yet accumulated. This observation aligns with our analysis in Fig. 4(b) of the paper. Moreover, both the middle and deep layers exhibit substantial and progressively compounded cross-modal interactions and language-induced bias, leading to stronger and more stable mitigation effects of AFTER.
>
>
>
> > ***Question 2:** The models used to generate training data are not at all mentioned.*
>
> Thank you for your suggestions. **We have mentioned and analyzed the three models used for generating training data  in Section 4.3, line 429~431**, namely GPT-4o (200B), GPT-4o-mini (8B), and LLaVA-v1.5 (7B). For clarity, we have now added an explicit statement at "Implementation Details" in Section 4.1.

---

> > ### Comment · Reviewer_n12p · 2025-11-25
> >
> > I have no further questions. Thanks for your reply. Scores are updated.

---

> > > ### Author Response · Authors · 2025-11-25
> > > **Acknowledgement to Reviewer n12p**
> > >
> > > Thank you very much for your time and for reconsidering our rebuttal. We will continue to refine and improve our work.

---

### Official Review · Reviewer_j2pv · 2025-10-31

**Soundness:** 3
**Presentation:** 3
**Contribution:** 3
**Rating:** 6
**Confidence:** 3

**Summary:**

This paper tackles hallucination by leaning on straightforward factual textual cues, rather than only nudging activations or poking the image with perturbations. The approach first uses dense ground-truth annotations from the COCO training set to build three fact banks (category, attribute, relation). These facts are turned into text descriptions with an LVLM. For each description, the method creates trusted and untrusted pairs to get positive and negative activation examples, which yield an editing vector. A small offset estimator is then trained for Query-Adaptive Offset optimization, giving a query-specific offset at inference. In practice, it reduced hallucinations and lifts accuracy across POPE, MME, and AMBER on various LVLM.

**Strengths:**

- The trusted vs. untrusted pairing is a smart way to bootstrap supervision without manual labeling at scale.
- Inference behaves like a lightweight adapter, so the method is efficient.
- Results are strong across both discriminative and generative benchmarks.
- Analysis is thorough, with useful ablations, including performance as the image pool grows from 50 to 500.
- The out-of-distribution check is informative and suggests the method does not degrade under domain or QA-style shifts (discriminative to generative).

**Weaknesses:**

- The pipeline relies on ground-truth object annotations during preparation, which may be unavailable in some domains. In QAO, query-focused supervision depends on extracting objects from the question and checking membership against the COCO-derived category fact set T_c; if a queried object is not in T_c, they synthesize a negative textual sub-description. Without rich annotations or a compatible taxonomy, this step becomes brittle and hard to port.
- Implementation detail: the method is model-specific, so you must recompute the steering vector and retrain the small offset estimator for each LVLM, which adds setup overhead and limits portability across architectures.

**Questions:**

- What explains the drop in Cover, the metric for answer comprehensiveness?
- Does the learned steering direction favor frequent classes at the expense of rare or novel ones?
- How does the method handle compositional prompts with multiple objects, attributes, and relations?

---

> ### Author Response · Authors · 2025-11-20
> **Response to Reviewer j2pv (1/2)**
>
> Thank you very much for supporting our methodology and giving valuable feedback.
>
> > ***Weakness 1:** The pipeline relies on ground-truth object annotations during preparation, which may be unavailable in some domains. In QAO, query-focused supervision depends on extracting objects from the question and checking membership against the COCO-derived category fact set T_c; if a queried object is not in T_c, they synthesize a negative textual sub-description. Without rich annotations or a compatible taxonomy, this step becomes brittle and hard to port?*
>
> Thank you for your meaningful comment. **In annotation-scarce scenarios,** it is not necessary to annotate large amounts of data through our pipeline. **Only a few densely annotated samples(e.g. 5 samples) are sufficient** for AFTER to still exhibit considerable performance. This property greatly **reduces both the need for and the difficulty of applying the pipeline in annotation-scarce settings**. More analysis of performance on annotation-scarce settings is illustrated in Appendix A.8 of our revised paper.
>
> **Moreover, AFTER also ensures the reliability of the query-focused supervision in two ways**:
>
> - **Following the query construction protocols of POPE[1] and AMBER[2],** we guarantee that any queried object is either: **(i) present in the GT annotations, or (ii) absent from both the GT annotations and the image context**.
> - We ensure that the **extracted object name $q_{i,j}$ strictly aligns with the GT label taxonomy** by prompting the model 𝐹 with the full list of GT labels, thereby avoiding mismatches caused by incompatible taxonomy.
>
> [1] Li Y, Du Y, Zhou K, et al. Evaluating Object Hallucination in Large Vision-Language Models[C]//Proceedings of the 2023 Conference on Empirical Methods in Natural Language Processing. 2023: 292-305.
>
> [2] Wang J, Wang Y, Xu G, et al. Amber: An llm-free multi-dimensional benchmark for mllms hallucination evaluation[J]. arXiv preprint arXiv:2311.07397, 2023.
>
> > ***Weakness 2:** Implementation detail: the method is model-specific, so you must recompute the steering vector and retrain the small offset estimator for each LVLM, which adds setup overhead and limits portability across architectures.*
>
> Thanks for your valuable feedback. In fact,**most hallucination mitigation techniques** (e.g. training-based SFT[1], DPO[2], RLHF[3], inference-based contrastive decoding[4][5], as well as activation editing methods[6-9]) **are also model-specific**, which require retraining or access to model-dependent information whenever the applied model changes. Compared with other methods, activation editing **incurs relatively low overhead when switching to a new model and therefore offers practical transferability**. We have further elaborated in Section 2.2 of our revised paper.
>
> In future work, we aim to explore lightweight modules to project editing vectors generated from one model into another model's feature space, thereby achieving more efficient cross-model vector transplantation.
>
> [1]Wang W, Ren Y, Luo H, et al. The all-seeing project v2: Towards general relation comprehension of the open world[C]//European Conference on Computer Vision. Cham: Springer Nature Switzerland, 2024: 471-490.
>
> [2] Ouali Y, Bulat A, Martinez B, et al. Clip-dpo: Vision-language models as a source of preference for fixing hallucinations in lvlms[C]//European Conference on Computer Vision. Cham: Springer Nature Switzerland, 2024: 395-413.
>
> [3] Yu T, Yao Y, Zhang H, et al. Rlhf-v: Towards trustworthy mllms via behavior alignment from fine-grained correctional human feedback[C]//Proceedings of the IEEE/CVF Conference on Computer Vision and Pattern Recognition. 2024: 13807-13816.
>
> [4] Zhang H, Chen H, Chen M, et al. Active layer-contrastive decoding reduces hallucination in large language model generation[C]//Proceedings of the 2025 Conference on Empirical Methods in Natural Language Processing. 2025: 3028-3046.
>
> [5] Zhang Y, Cui L, Shi S. Alleviating hallucinations of large language models through induced hallucinations[C]//Findings of the Association for Computational Linguistics: NAACL 2025. 2025: 8218-8232.
>
> [6] Liu S, Ye H, Zou J. Reducing hallucinations in large vision-language models via latent space steering[C]//The Thirteenth International Conference on Learning Representations. 2025.
>
> [7]Chen J, Zhang T, Huang S, et al. Ict: Image-object cross-level trusted intervention for mitigating object hallucination in large vision-language models[C]//Proceedings of the Computer Vision and Pattern Recognition Conference. 2025: 4209-4221.
>
> [8] Li K, Patel O, Viégas F, et al. Inference-time intervention: Eliciting truthful answers from a language model[J]. Advances in Neural Information Processing Systems, 2023, 36: 41451-41530.
>
> [9] Wang T, Jiao X, Zhu Y, et al. Adaptive activation steering: A tuning-free llm truthfulness improvement method for diverse hallucinations categories[C]//Proceedings of the ACM on Web Conference 2025. 2025: 2562-2578.

---

> ### Author Response · Authors · 2025-11-20
> **Response to Reviewer j2pv (2/2)**
>
> > ***Question 1:** What explains the drop in Cover, the metric for answer comprehensiveness?*
>
> Thank you for your valuable feedback. Consistent with our results, **many hallucination mitigation studies [1][2][3] also ****observe a drop in the Cover metric**,** and explain this phenomenon as a“trade-off between factual completeness and hallucination suppression”**[2]. Our analyses attribute this effect to the tendency of hallucination-reduction techniques to drive the LVLM toward producing shorter outputs. As an important future direction, we plan to investigate an activation editing strategy that simultaneously improves coverage while reducing hallucinations.
>
> [1] Compagnoni A, Caffagni D, Moratelli N, et al. Mitigating hallucinations in multimodal llms via object-aware preference optimization[J]. arXiv preprint arXiv:2508.20181, 2025.
>
> [2] Han M, Hao H, Zhou J, et al. Self-Consistency as a Free Lunch: Reducing Hallucinations in Vision-Language Models via Self-Reflection[J]. arXiv preprint arXiv:2509.23236, 2025.
>
> [3] Guo Z, Man X, Xu H, et al. LISA: A Layer-wise Integration and Suppression Approach for Hallucination Mitigation in Multimodal Large Language Models[J]. arXiv preprint arXiv:2507.19110, 2025.
>
> > ***Question 2:** Does the learned steering direction favor frequent classes at the expense of rare or novel ones?*
>
> Thank you for your constructive comments. **The steering direction learned by AFTER is class-agnostic and general, without favouring any classes.** It represents the comprehensive associations between textual and visual information within LVLMs.
>
> To further illustrate this, we evaluate the steering direction trained on COCO's 80 base classes using two subsets of GQA: (i) the subset corresponding to the same 80 base classes, and (ii) a subset of 47 novel classes sourced from YFCC100M [1] (e.g.,*rock, door, railroad*) that do not appear in COCO. This setup simulates the experimental configuration for frequent versus rare classes. The separate accuracy scores are included in Appendix A.9 of the revised paper and reported as follows:
>
> |||||||
> |-|-|-|-|-|-|
> |Classes  |Methods  |Models||||
> |||LLaVA-v1.5-7B|InstructBlip|Shikra|Qwen2.5-VL|
> |Base classes in COCO|Baseline|89.5|88|84.6|87.9|
> ||Ours|**92.3（+2.8）**|**89.5（+1.5）**  |**88.7(+4.1)**|**90.1(+2.2)**|
> |Novel classes in GQA|Baseline|66.7|71.9|74.4|80|
> ||Ours|**78.5（+11.8）**|**76.2（+4.3）**|**77.2（+2.8）**|**83.8(+3.8)**|
>
> The results demonstrate that AFTER achieves even higher ACC improvements on novel classes than on previously seen base classes. This can be attributed to the combination of lower baseline performance on novel classes and the inherently category-agnostic nature of our factual-guided steering vector. With a higher frequency of hallucinations, the category-agnostic AFTER is able to achieve greater improvements.
>
> [1]Thomee B, Shamma D A, Friedland G, et al. Yfcc100m: The new data in multimedia research[J]. Communications of the ACM, 2016, 59(2): 64-73.
>
> > ***Question 3:** How does the method handle compositional prompts with multiple objects, attributes, and relations?*
>
> Thank you for your valuable feedback. **The steering vector learned by AFTER is derived from a combined factual guidance of multiple objects, attributes, and relationships, therefore demonstrating notable capability in handling such compositional prompts**.
>
> To illustrate this, we evaluate AFTER on the original GQA dataset[1], which is specifically designed for compositional question answering that containing multiple objects, attributes, and relationships (e.g. *"What color is the fruit on the right side, red or green?"* ), to test its ability to handle compositional prompts. Following the official implementation of LLaVA [2], we conduct evaluation on the testdev_balanced set and report ACC metrics on LLaVA-v1.5. The results are included in Appendix A.10 of the revised paper and listes as follows:
>
> |||
> |-|-|
> |Methods|ACC|
> |Baseline|61.6|
> |ICT|61.7(+0.1)|
> |Ours|63.9(+2.3)|
>
>
> It is evident that AFTER consistently outperforms both the baseline and the SOTA activation editing method ICT, demonstrating its effectiveness in compositional reasoning and its ability to leverage the combined object-attribute-relation information in steering the LVLM's behavior.
>
> [1]Hudson D A, Manning C D. Gqa: A new dataset for real-world visual reasoning and compositional question answering[C]//Proceedings of the IEEE/CVF conference on computer vision and pattern recognition. 2019: 6700-6709.
>
> [2] https://github.com/haotian-liu/LLaVA

---

> ### Author Response · Authors · 2025-11-26
> **Follow-up with Reviewer j2pv**
>
> Dear Reviewer j2pv,
>
> Thank you for handling our manuscript and providing valuable feedback. We hope that our responses have sufficiently addressed the concerns you raised. We welcome more discussion if you have more questions and suggestions. As the discussion deadline is approaching, we would be very grateful if you could take a moment to review our reply.

---

### Official Review · Reviewer_cVcN · 2025-10-31

**Soundness:** 2
**Presentation:** 4
**Contribution:** 2
**Rating:** 4
**Confidence:** 3

**Summary:**

This paper addresses the issue of object hallucination in Large Vision-Language Models (LVLMs).The authors think  that they arises mainly from language bias and manifests as category, attribute, or relation errors. The authors propose AFTER (Adaptive Factual-guided Visual-Textual Editing for hallucination mitigation), a lightweight inference-time activation editing method. AFTER combines Factual-Augmented Activation Steering (FAS), which transforms ground-truth visual annotations into textual facts to provide positive, factual guidance, and Query-Adaptive Offset Optimization (QAO), which learns query-specific adjustments to better align visual-textual associations.

**Strengths:**

1. The figures and illustrations are excellent — the visuals are carefully designed and aesthetically pleasing, and they greatly help in clarifying certain issues I encountered during reading.
2. The writing is strong, and the appendix provides abundant experimental details, which convinces me that the reported results can be reproduced directly based on the information given in the paper.
3. The motivation is solid and well-founded. Similar concerns have been raised in other works, and the paper rightly emphasizes that language bias may lead LVLMs to overlook important information in the images [1].

[1] Jia H, Jiang C, Xu H, et al. Symdpo: Boosting in-context learning of large multimodal models with symbol demonstration direct preference optimization[C]//Proceedings of the Computer Vision and Pattern Recognition Conference. 2025: 9361-9371.

**Weaknesses:**

1. The proposed method is evaluated primarily on relatively simple datasets, whose difficulty appears notably lower than the complexity illustrated in Figure 1. It remains unclear whether the approach would retain its effectiveness on more challenging benchmarks, such as HallusionBench[1] or CRPE[2].
2. The approach relies heavily on datasets like COCO, which are richly annotated by humans, and thus depends on the availability of high-quality manual annotations. Given that the current results are derived from COCO-like data, it is uncertain whether comparable performance could be achieved on domains where such annotations are scarce or costly—e.g., in medical imaging datasets.
3. Regarding datasets such as BLINK[3], it is unclear whether the method could deliver strong generalization performance. In the current generalization experiments, COCO is evidently an in-domain dataset, making it difficult to assess how well the method would transfer to truly out-of-domain scenarios.

[1] Guan T, Liu F, Wu X, et al. Hallusionbench: an advanced diagnostic suite for entangled language hallucination and visual illusion in large vision-language models[C]//Proceedings of the IEEE/CVF Conference on Computer Vision and Pattern Recognition. 2024: 14375-14385.

[2] Wang W, Ren Y, Luo H, et al. The all-seeing project v2: Towards general relation comprehension of the open world[C]//European Conference on Computer Vision. Cham: Springer Nature Switzerland, 2024: 471-490.

[3] Fu X, Hu Y, Li B, et al. Blink: Multimodal large language models can see but not perceive[C]//European Conference on Computer Vision. Cham: Springer Nature Switzerland, 2024: 148-166.

**Questions:**

As described in Weakness.

---

> ### Author Response · Authors · 2025-11-20
> **Response to Reviewer cVcN(1/3)**
>
> We greatly appreciate your constructive suggestions and favorable comments on AFTER's motivation and presentation.
>
> > ***Weakness 1:** The proposed method is evaluated primarily on relatively simple datasets, whose difficulty appears notably lower than the complexity illustrated in Figure 1. It remains unclear whether the approach would retain its effectiveness on more challenging benchmarks, such as HallusionBench[1] or CRPE[2].*
>
> Thank you for your valuable comments. **Following previous works[1-4], our evaluation datasets are the most commonly adopted and representative benchmarks in prior hallucination mitigation research**. Strong performance on these datasets effectively demonstrates the method's effectiveness.
>
> **Additional experiments on more challenging benchmark CRPE demonstrate that AFTER still retains its effectiveness**. Specifically, we conducted experiments on CRPE[5] under the same experimental settings as POPE, and have included the corresponding results in Appendix A.6 of the revised manuscript. The results are as follows:
>
> |||||||
> |-|-|-|-|-|-|
> |Methods  |Existence|Relation||||
> |||Subject|Predicate|Object|Overall|
> |Baseline|82.8|50.2|55.2|64.0|56.5|
> |Ours|84.9(+2.1)|51.8(+1.6)|57.8(+2.6)|67.6(+3.6)|59.1(+2.6)|
>
>
> The above results show that AFTER achieves consistent improvements over the baseline across all evaluated dimensions, with notable 3.6% gains of Relation task in the Object dimension. This indicates that our method can still accurately captures the presence of objects and their relationships in a more challenging context.
>
> [1] Liu S, Ye H, Zou J. Reducing hallucinations in large vision-language models via latent space steering[C]//The Thirteenth International Conference on Learning Representations. 2025.
>
> [2]Chen J, Zhang T, Huang S, et al. Ict: Image-object cross-level trusted intervention for mitigating object hallucination in large vision-language models[C]//Proceedings of the Computer Vision and Pattern Recognition Conference. 2025: 4209-4221.
>
> [3]Leng S, Zhang H, Chen G, et al. Mitigating object hallucinations in large vision-language models through visual contrastive decoding[C]//Proceedings of the IEEE/CVF Conference on Computer Vision and Pattern Recognition. 2024: 13872-13882.
>
> [4]Huang Q, Dong X, Zhang P, et al. Opera: Alleviating hallucination in multi-modal large language models via over-trust penalty and retrospection-allocation[C]//Proceedings of the IEEE/CVF Conference on Computer Vision and Pattern Recognition. 2024: 13418-13427.
> [5]Wang W, Ren Y, Luo H, et al. The all-seeing project v2: Towards general relation comprehension of the open world[C]//European Conference on Computer Vision. Cham: Springer Nature Switzerland, 2024: 471-490.

---

> ### Author Response · Authors · 2025-11-20
> **Response to Reviewer cVcN(2/3)**
>
> > ***Weakness 2:** The approach relies heavily on datasets like COCO, which are richly annotated by humans, and thus depends on the availability of high-quality manual annotations. Given that the current results are derived from COCO-like data, it is uncertain whether comparable performance could be achieved on domains where such annotations are scarce or costly—e.g., in medical imaging datasets.*
>
>  Thank you for your meaningful comments. **Demonstrated by previous activation editing methods[1][2], our method exhibits considerable peformance with few densely annotated samples(e.g. 5 samples)**, **as the steering vector is easy to identify.** **This property enables our method to perform effectively even in annotation-scarce settings****.** To empirically validate this capability, we extend the training-size analysis presented in Appendix A.1 to include experiments with 5, 10, and 30 training samples. The results are shown in the revised Figure 5. Notably, even when using only 5 samples to construct factual-guided editing vectors and train QAO, our method achieves ACC and F1 scores of 81.0% and 82.1%, outperforming the baseline LLaVA-v1.5 by 1.8% and 1.2%.
>
> **Experiments on medical imaging benchmark SLAKE[3] demonstrate that AFTER achieves notable performance** **in domains where high-quality annotations are scarce or costly.** Specifically, SLAKE is a well-adopted medical VQA benchmark that provides bounding-box annotations for organs and diseases, making it suitable for generating factual textual descriptions using AFTER. **To mimic the real-world annotation-scarce scenario,** we follow the same experimental setup and construct editing vectors using 5, 10, 30, and 50 samples. The comprehensive results using the medical LVLM LLaVA-med-v1.5[4] are appended to Appendix A.7 in the revised version, and reported in the following table:
>
> ||||
> |-|-|-|
> |Methods||ACC|
> |Baseline||50.3|
> |Ours    |5 samples|51.3|
> ||10 samples|51.6|
> ||30 samples|51.9|
> ||50 samples|**52.0**|
>
>
> We can also observe an improvement of at least 1.0% in ACC when using only 5 samples, reinforcing the feasibility and effectiveness of our method in annotation-scarce domains.
>
> [1] Li K, Patel O, Viégas F, et al. Inference-time intervention: Eliciting truthful answers from a language model[J]. Advances in Neural Information Processing Systems, 2023, 36: 41451-41530.
>
> [2] Wang T, Jiao X, Zhu Y, et al. Adaptive activation steering: A tuning-free llm truthfulness improvement method for diverse hallucinations categories[C]//Proceedings of the ACM on Web Conference 2025. 2025: 2562-2578.
>
> [3]Liu B, Zhan L M, Xu L, et al. Slake: A semantically-labeled knowledge-enhanced dataset for medical visual question answering[C]//2021 IEEE 18th international symposium on biomedical imaging (ISBI). IEEE, 2021: 1650-1654.
>
> [4]Li C, Wong C, Zhang S, et al. Llava-med: Training a large language-and-vision assistant for biomedicine in one day[J]. Advances in Neural Information Processing Systems, 2023, 36: 28541-28564.

---

> ### Author Response · Authors · 2025-11-20
> **Response to Reviewer cVcN(3/3)**
>
> > ***Weakness 3:** Regarding datasets such as BLINK[3], it is unclear whether the method could deliver strong generalization performance. In the current generalization experiments, COCO is evidently an in-domain dataset, making it difficult to assess how well the method would transfer to truly out-of-domain scenarios.*
>
> Thanks for your insightful feedback. **We follow ICT[1], which treats COCO and GQA as "different distributions", and therefore conduct a generalization experiment of COCO→GQA**. The rationale is that although the image domains of COCO and GQA are visually similar, their category spaces differ substantially: GQA [2] not only contains COCO's 80 categories but also includes many additional categories drawn from YFCC100M[3]. To quantify this divergence, we analyzed the 9,000 questions across three GQA-based settings from the official repository of POPE [4], and found that **5,240 (58.2%)** questions involve **47 novel categories** that are not present in COCO (for example,*rock*,*door*,*railroad*, etc.). This indicates that COCO→GQA generalization experiments can evaluate AFTER's out-of-domain generalizability to some extent.
>
> **Our further COCO→BLINK generalization experiments demonstrate that we still achieve notable generalization performance to truly out-of-domain scenarios.** Specifically, we evaluate cross-domain generalization with Qwen2.5-vl-8B-Instruct by applying editing vectors derived from COCO to the out-of-domain BLINK dataset. The experiments are shown in Appendix A.8 of the revised paper, with the results presented in the following table:
>
> |||||||||||||||||
> |-|-|-|-|-|-|-|-|-|-|-|-|-|-|-|-|
> |Methods|**Averaged**|OL|RR  |C  |RD|VC|AS|FC|IT|MR |SC|VS|FD|J|SR|
> |Baseline|54.5|51.6|38.8|69.2|64.5|54.1|**63.3**|**23.1**|26.0|**46.6**|34.5|88.9|46.2|**66.7**|**89.5**|
> |Ours|**55.7**|**56.9**|**40.3**|**73.6**|**66.2**|**54.6**|63.1|22.9|**27.2**|46.5|**35.9**|**89.0**|**47.6**|66.3|**89.5**|
>
>
> The results show that, relative to the baseline, our method achieves an average improvement of 1.2%, with particularly notable gains on tasks such as Object Localization and Counting. These findings demonstrate that our approach retains strong generalization ability even when applied to out-of-domain image sources. It is worth noting that the generalization performance of AFTER on **multi-image tasks** (e.g., Functional Correspondence(FC):–0.2%, Visual Correspondence(VC):+0.5%, Multi-view Reasoning(MR):–0.1%, among 8 tasks) is notably weaker than on **single-image tasks** (including **Object Localization(OL):+5.3%, Counting(C):+4.4%, Relative Depth(RD):+2.3%**, among 6 tasks). We conjecture that this may stem from the substantial discrepancy between multi-image and single-image input paradigms.
>
> [1]Chen J, Zhang T, Huang S, et al. Ict: Image-object cross-level trusted intervention for mitigating object hallucination in large vision-language models[C]//Proceedings of the Computer Vision and Pattern Recognition Conference. 2025: 4209-4221.
>
> [2]Hudson D A, Manning C D. Gqa: A new dataset for real-world visual reasoning and compositional question answering[C]//Proceedings of the IEEE/CVF conference on computer vision and pattern recognition. 2019: 6700-6709.
>
> [3]Thomee B, Shamma D A, Friedland G, et al. Yfcc100m: The new data in multimedia research[J]. Communications of the ACM, 2016, 59(2): 64-73.
>
> [4]https://github.com/RUCAIBox/POPE

---

> ### Author Response · Authors · 2025-11-26
> **Follow-up with Reviewer cVcN**
>
> Dear Reviewer cVcN,
>
> Thank you for handling our manuscript and providing valuable feedback. We hope that our responses have sufficiently addressed the concerns you raised. We welcome more discussion if you have more questions and suggestions. As the discussion deadline is approaching, we would be very grateful if you could take a moment to review our reply.

---

### Official Review · Reviewer_dssQ · 2025-10-31

**Soundness:** 3
**Presentation:** 3
**Contribution:** 3
**Rating:** 6
**Confidence:** 4

**Summary:**

The paper presents AFTER to mitigate object hallucinations in LVLMs. These hallucinations, arising from language bias, are categorized into three types: category, attribute, and relation hallucinations. AFTER address these hallucinations during inference by adaptively steering visual-textual activations toward factual semantic guidance. The paper evaluates the effectiveness of AFTER on several benchmarks, including POPE and MME, where it demonstrates superior performance over baseline methods, e.g., achieving up to a 16.3% reduction in hallucination.

**Strengths:**

AFTER stands out by combining factual textual guidance with query-specific offsets to improve visual-textual activation editing. FAS uses factual annotations from images to provide clear guidance, effectively reducing language bias. Meanwhile, QAO adapts the editing process by creating query-specific offsets, allowing the model to handle distinct visual-textual associations more effectively. This innovation overcomes the limitation of previous methods that often use a single, averaged editing vector.

**Weaknesses:**

- AFTER relies on accessing activations from open-source LLMs, which limits its applicability to closed-source models. This restricts the method’s use in private large-scale models or those that are not publicly accessible. Please explore post-processing steps based on model outputs during inference
- QAO is presented as a key component to improve editing diversity and accuracy, the paper lacks in-depth analysis of its training process and its specific impact on performance, particularly in handling different queries. Visualizing the offsets for different queries could provide a clearer understanding of how QAO enhances the editing process.
- The paper mentions tuning the editing strength and the number of edited heads based on empirical results.

**Questions:**

The method uses textual descriptions derived from ground-truth annotations to steer the activation. What is the impact of varying the quality or diversity of the factual textual descriptions (t+) on the hallucination mitigation process? Is there a specific threshold or methodology for ensuring that these descriptions remain reliable across different domains?

---

> ### Author Response · Authors · 2025-11-20
> **Response to Reviewer dssQ (1/2)**
>
> We sincerely thank you for your detailed review and positive feedback on our innovation.
>
> > ***Weakness 1**: AFTER relies on accessing activations from open-source LLMs, which limits its applicability to closed-source model. This restricts the method’s use in private large-scale models or those that are not publicly accessible. Please explore post-processing steps based on model outputs during inference.*
>
> Thank you for your meaningful comment. **Not only AFTER, all activation editing methods [1–3] require editing internal activations of LLMs to calibrate their outputs.** Therefore, they are inherently tailored for open-source LLMs. We have stated it in Section 5, Conclusion.
>
> Although post-processing steps based on model outputs do not fall under the scope of activation editing, we can nonetheless extend the factual-guided philosophy of AFTER from activation space editing to model output editing. For example, we can explore output-level factual-guided self-consistency regeneration to enable hallucination reduction without requiring access to internal representations.
>
> [1] Liu S, Ye H, Zou J. Reducing hallucinations in large vision-language models via latent space steering[C]//The Thirteenth International Conference on Learning Representations. 2025.
>
> [2]Chen J, Zhang T, Huang S, et al. Ict: Image-object cross-level trusted intervention for mitigating object hallucination in large vision-language models[C]//Proceedings of the Computer Vision and Pattern Recognition Conference. 2025: 4209-4221.
>
> [3] Li K, Patel O, Viégas F, et al. Inference-time intervention: Eliciting truthful answers from a language model[J]. Advances in Neural Information Processing Systems, 2023, 36: 41451-41530.
>
> > ***Weekness 2:** QAO is presented as a key component to improve editing diversity and accuracy, the paper lacks in-depth analysis of its training process and its specific impact on performance, particularly in handling different queries. Visualizing the offsets for different queries could provide a clearer understanding of how QAO enhances the editing process.*
>
> Thank you for your constructive comments. **Our in-depth analysis of QAO demonstrates its ability to generate query-specific offsets under the guidance of fine-grained factual semantics, thereby effectively addressing diverse language bias**.
>
> **We first provide t-SNE and PCA-reduced visualizations of offsets to illustrate how QAO adapts to different queries and thereby enhances the editing process**. Specifically, we examine：(i) the t-SNE visualization of different queries' editing vectors after adding the offsets. (ii) the PCA-reduced 1D feature for different query activations edited with and without offsets, and compare against the distribution of factual activations. These visualizations are presented in Figure 7 and Appendix A.4 of the revised manuscript.
>
> As shown in the right-hand t-SNE plot, the query-specific vectors produced by QAO form a diverse distribution around the general vector $\bar{\mathbf{d}}$. Prior methods that rely solely on $\bar{\mathbf{d}}$ necessarily lose query-specific information and thus cannot achieve fine-grained editing. In contrast, **the QAO fully captures query-relevant visual semantics to generate offsets varying meaningfully in both direction and magnitude, enabling the fine-grained correction of language bias**. As illustrated in the two left-hand plots, applying the QAO-generated query-specific offset consistently shifts the model's activations **closer to the factual textual activation, which represents the activation space of LVLM to produce factually correct outputs**. This demonstrates that QAO can adaptively steer the edited activations to the activation space, thereby substantially improving hallucination mitigation.
>
> **We further compare different offset supervision strategies for training QAO, and include the results in Appendix A.4**. Specifically, the compared variants include: 1) **Fixed strategy** that uses a constant offset. 2) **Random strategy** that uses a Gaussian-sampled offset. 3) **$t^+$-guided strategy** that training QAO offset with $t^+$, and 4) **$t^\*$ guided strategy** that training QAO offset with $t^*$. The comparative results are summarized in the following table:
>
> |||||
> |-|-|-|-|
> |Methods||POPE(ACC↑) |AMBER(Hal↓)|
> |w/o QAO||83.8|22.3|
> |w QAO|Fixed |81.6|28.9|
> ||Random|83.2|23.6|
> ||$t^+$ guided|84.9|20.9|
> ||$t^*$ guided|**85.7**|**20.5**|
>
> We observe that fixed or random strategies fail to provide consistent improvements and can even degrade performance, indicating that factual description guidance is essential for training QAO to generate the meaningful, semantically aligned offsets. Moreover, compared with complete description $t^+$ guided offset, the query-focused offset achieves the best performance on both POPE and AMBER, confirming that QAO effectively maximizes both the diversity and the efficacy of activation editing by leveraging our constructed diverse, query-relevant semantics $t^*$.

---

> ### Author Response · Authors · 2025-11-20
> **Response to Reviewer dssQ (2/2)**
>
> > ***Weakness 3:** The paper mentions tuning the editing strength and the number of edited heads based on empirical results.*
>
> Thank you for your valuable comments.**Due to the uninterpretable nature of LVLMs, all activation editing methods [1-3] cannot specify the optimal editing strength or the appropriate head num without an established theory,** and therefore **rely on empirical hyperparameter tuning**. In practice, AFTER can **maintain stable performance on new datasets when initialized with the same prior hyperparameters. Only minor adjustments are required to reach optimal performance**. We have further clarified in Appendix C.5 of our revised paper.
>
> To illustrate this, we take baseline Shikra as an example and analyze the optimal values of α and K across three datasets, with the results summarized in the table below.
> |||||
> |-|-|-|-|
> ||POPE (ACC↑)|MME (E↑)|AMBER (CH↓)|
> |**Optimal value of (α,K)**|(9, 64)|(9, 64)|(9, 72)|
> |**Performance when α=9，K=64**|82.5|190|7.1|
> |**Optimal Performance**|**82.5（+0.0）**|**190（+0.0）**|**6.9（-0.2）**|
>
> Experimental findings show that the optimal values of (α,K) is (9, 64) for both POPE and MME . Further evaluation reveals that  with (α,K)=(9, 64), the CHAIR metric on AMBER is only 0.2 higher than the optimal result, a negligible difference. Therefore, when transferring AFTER to new tasks, we can directly apply the empirically derived hyperparameters (*e.g.* (α,K)=(9, 64) for Shikra) to achieve considerable editing effects.
>
> [1]Chen J, Zhang T, Huang S, et al. Ict: Image-object cross-level trusted intervention for mitigating object hallucination in large vision-language models[C]//Proceedings of the Computer Vision and Pattern Recognition Conference. 2025: 4209-4221.
> [2] Li K, Patel O, Viégas F, et al. Inference-time intervention: Eliciting truthful answers from a language model[J]. Advances in Neural Information Processing Systems, 2023, 36: 41451-41530.
> [3] Wang T, Jiao X, Zhu Y, et al. Adaptive activation steering: A tuning-free llm truthfulness improvement method for diverse hallucinations categories[C]//Proceedings of the ACM on Web Conference 2025. 2025: 2562-2578.
>
> > ***Question 1:** The method uses textual descriptions derived from ground-truth annotations to steer the activation. What is the impact of varying the quality or diversity of the factual textual descriptions (t+) on the hallucination mitigation process? Is there a specific threshold or methodology for ensuring that these descriptions remain reliable across different domains?*
>
> Thank you for your valuable feedback. We will answer your concerns from following two aspects:
>
> 1. **The impact of varying the quality or diversity of t+**
>
> The quality and diversity are two critical properties for t+, both of which are ensured under our construction setting.
>
> - **In terms of quality, our construction process of t+ ensures that the ground-truth information is neither distorted nor lost, preventing the introduction of hallucinations or other low-quality content**. Consequently, t+ maintains high quality even in annotation-scarce domains. Only in cases where the ground-truth contains substantial errors (such as due to annotators' limited expertise or overly complex data) might generate low-quality t+
> - **In terms of diversity, our experiment shows that diversity of t+ has minimal impact on performance**. Specifically, we design a controlled experiment, and append the results and analysis to Appendix A.5 in our revised paper. We prompt GPT-4o to remove all non-essential diversified information apart from GT from the original high-quality t+, producing low-diversity yet factually accurate descriptions. The results are presented in the table below, where "Direct Input" denotes using t+ directly as model input (indirectly reflecting the quality of t+) while“Steering Vector” reflects the hallucination-mitigation effect of AFTER:
>
> ||||||
> |-|-|-|-|-|
> |Different diversity of t+|Direct Input||Steering Vector||
> ||ACC|F1|ACC|F1|
> |low-diversity|93.6|93.9|85.3|84.4|
> |high-diversity|93.9|93.7|85.3|84.5|
>
> As shown in the table, when factuality is held constant, high-diversity and low-diversity t+ yield nearly identical results.
>
> *2. **Methods for ensuring the reliability of t+***
>
> **We can ensure the reliability of t+ by applying a prior threshold-based constraint.** Specifically, we can assess the **reliability of t+ by directly using it as model input and comparing its resulting accuracy** against a predefined reliability threshold. For a given task, the **threshold can be determined by comparing the editing effect t+ with the baseline performance**. For instance, in experiments on the COCO dataset, we observe that a threshold σ = 70% is sufficient for t+to provide reliable guidance. In practice, the t+ used in our experiments consistently maintains high quality. If the accuracy falls below σ, the corresponding t+ requires additional refinement, such as manual verification or correction, to ensure sufficient reliability.

---

> > ### Comment · Reviewer_dssQ · 2025-11-25
> >
> > Thank you for the response. Most of my concerns have been addressed. I will keep my score.

---

> > > ### Author Response · Authors · 2025-11-25
> > > **Acknowledgment to Reviewer dssQ**
> > >
> > > Thank you very much for taking the time to review our rebuttal. If you have any further questions, please feel free to contact us at any time.

---

### Author Response · Authors · 2025-11-20
**Response to All ACs and Reviewers**

We thank all the ACs and reviewers for their thoughtful and valuable feedback! We have conducted additional experiments and revised the paper based on the reviews. We highlighted all changes in red. We also included more results and analysis to make the paper more comprehensive. Below is a summary of the main changes. Please let us know if you have further questions.

|||||
|-|-|-|-|
|**Reviewer**|**Questions**|**Section & Paragraph**|**Change**|
|Reviewer dssQ|Weakness 1|Section 5|Clarifying the open-source property of activation editing|
||Weakness 2|Appendix A.4|Adding more analysis of QAO|
||Weakness 3|Appendix C.5|Clarifying the hyperparameter tuning |
||Question 1|Appendix A.5|Adding more analysis on Diversity of t+|
|Reviewer cVcN|Weakness 1|Appendix A.6|Adding experiments on more challenging benchmark CRPE|
||Weakness 2|Appendix A.1|Extending experiments with 5, 10, 30 samples|
|||Appendix A.7|Adding experiments on data-scarce scenario SLAKE|
||Weakness 3|Appendix A.8|Adding more experiments to evaluate generalization on BLINK|
|Reviewer j2pv|Weakness 1|Appendix A.7|Adding experiments on data-scarce scenario SLAKE|
||Weakness 2|Section 2.2|Clarifying the advantage of activation editing among model-specific methods|
||Question 1|Section 4.2——"Hallucination Mitigation Performance"|Analysing the Cover metric|
||Question 2|Appendix A.9|Adding more experiments on novel class|
||Question 3|Appendix A.10|Adding more experiments on compositional prompts|
|Reviewer n12p|Weakness 1, Weakness 2|Section 4.1——"Implementation Details "|Detailing the construction of t+ and questions|
||Weakness 3|Appendix A.11|Adding more comparison experiments on more models and baselines|
||Weakness 4, Weakness 5|Section 3.3|Correcting and clarifing the symbol and equation|
||Weakness 6|Appendix A.8|Adding more experiments to evaluate generalization on BLINK|
||Question 1|Appendix A.12|Adding more analysis of different editing layers|
||Question 2|Section 4.1——"Implementation Details "|Detailing the construction of t+ and questions|

---

### Meta-Review · Area_Chair_934o · 2026-01-08

**Summary:**

This paper received 4 reviews. The reviewers (score/confidence) are: `dssQ (6/4), n12p (4/4), cVcN (4/3), j2pv (6/3)`.

Their major concerns:

Methodology:

- Key component QAO lacks in-depth training process and performance impact analysis; FAS has unclear question set construction logic and unvalidated synthesized textual descriptions (t+) (dssQ (6/4), n12p (4/4)).


Experiments:
- Evaluations are based on simple datasets; effectiveness on challenging benchmarks (e.g., HallusionBench, CRPE) is unproven (cVcN (4/3)).
- Compared models/baselines are outdated (all 2024) for a late 2025 submission; training and benchmark datasets share MSCOCO images, failing to prove cross-domain generalization (n12p (4/4)).

**Reviewer Concerns:**

Two negative reviewers:

- Reviewer n12p has acknowledged that the concerns have been addressed. The score is raised from 4 to 6.

- Reviewer cVcN raised a few concerns, mostly about the generalization ability (e.g., to weakly annotated data such as medical data). The authors have provided sufficient empirical evidence during rebuttal. The reviewer did not respond eventually. Based on the discussions, I believe the concerns of this reviewer should be addressed by the rebuttal.

**Reviewer Scores:**

Two negative reviewers:

- Reviewer n12p has acknowledged that the concerns have been addressed. The score raised from 4 to 6.

- Reviewer cVcN's concerns should be addressed by the rebuttal. I think the reviewer will raise the score.

---

### Decision · Program_Chairs · 2026-01-26

Accept (Poster)